



# Effects of surface water interactions with karst groundwater on microbial biomass, metabolism, and production

Adrian Barry-Sosa[1], Madison K. Flint[2], Justin C. Ellena[1], Jonathan B. Martin[2], Brent C. Christner[1]

[1]Department of Microbiology and Cell Science, University of Florida, Gainesville, 32611, USA
[2]Department of Geological Sciences, University of Florida, Gainesville, 32611, USA

*Correspondence to*: Brent C. Christner (xner@ufl.edu)

**Abstract.** Unearthing the effects of surface water and groundwater interactions on subsurface biogeochemical reactions is crucial for developing a more mechanistic understanding of carbon and energy flow in aquifer ecosystems. To examine physiological characteristics across groundwater microbial communities that experience varying degrees of interaction with

surface waters, we investigated ten springs and a river sink and rise system in North Central Florida that discharge from and/or mix with the karstic Upper Floridan Aquifer (UFA). Groundwater with longer residence times in the aquifer had lower concentrations of dissolved oxygen, dissolved and particulate organic carbon, and microbial biomass, as well as the lowest rates of respiration (0.102 to 0.189 mg $O_2$ $L^{-1}$ $d^{-1}$) and heterotrophic production (198 to 576 µg C $L^{-1}$ $d^{-1}$). Despite these features, oligotrophic UFA groundwater (< 0.5 mg C $L^{-1}$) contained bioavailable organic matter that supported

doubling times (14 to 62 h) and cell specific production rates (0.0485 to 0.261 pmol C $cell^{-1}$ $h^{-1}$) comparable to those observed for surface waters (17 to 20 h; 0.105 to 0.124 pmol C $cell^{-1}$ $h^{-1}$). The relatively high specific rates of dissimilatory and assimilatory metabolism indicate a subsurface source of labile carbon to the groundwater (e.g., secondary production and/or chemoautotrophy). Our results link variations in UFA hydrobiogeochemistry to the physiology of its groundwater communities, providing a basis to develop new hypotheses related to microbial carbon cycling, trophic hierarchy, and

processes generating bioavailable organic matter in karstic aquifer ecosystems.

## 1 Introduction

Groundwater is a vital natural resource that sustains aquatic ecosystems and provides approximately half of the water used globally for agriculture and human consumption (Jasechko and Perrone, 2021). Information on microbial biogeochemical reactions that affect organic matter degradation, nutrient cycling, and the transformation of contaminants in aquifers is

therefore highly relevant for understanding the processes contributing to groundwater quality. Moreover, knowledge of groundwater food webs is necessary for enabling meaningful assessments of their resilience to human impacts and environmental change. Aquifers vary in lithology and characteristics such as permeability, depth, and water storage capacity. Karst terrain covers ~15% of Earth's ice-free land surface (Goldscheider et al., 2020), and the dissolution of the underlying carbonate rock creates preferential flow paths for groundwater in the subsurface. More specifically, the solubility of karstic

rocks creates heterogenous subsurface environments with a range of permeabilities that can be described via a triple porosity model (i.e., rock's primary porosity; fractures; and conduits from a few cm to tens of m in diameter; Worthington et al.,





2000). Because the hydrogeological features of karst aquifers provide opportunities for rapid and direct exchange between surface waters and groundwater, they are extremely vulnerable to surface contaminants (e.g., Kalhor et al., 2019).

Studies of karstic aquifers from around the world have shown their groundwaters to be generally oligotrophic ($< 0.5$ mg C L$^{-1}$) and contain low standing stocks of microbial cells and biomass (Farnleitner et al., 2005; Wilhartitz et al., 2009, 2013; Hershey et al., 2018; Hershey and Barton, 2018; Malki et al., 2020, 2021). While these properties imply energy- and growth-limited conditions, new observations from karst aquifers are needed to decipher the biogeochemical contributions and rates of carbon metabolism by microbes in their groundwaters. Supplies of organic matter for subsurface microbial activity

include photosynthetically derived material originating from the surface (e.g. Jin, Jin et al., 2014) and that produced in situ via chemosynthesis. The current paradigm for karstic aquifers is that biogeochemical reactions are chiefly driven by the oxidation of surface-derived organic matter (Hershey and Barton, 2018). Based on this assumption, rates of carbon cycling and microbial growth should depend on the nature of the organic matter pool, presence of suitable electron acceptors, and residence time of groundwater. Though secondary production (i.e., heterotrophic production of biomass) may be the

dominant component of subsurface carbon and energy flow, the data available to assess rates of organic carbon recycling and heterotrophic growth in karst aquifer ecosystems are sparse. Remarkably, studies that have compared the chemistry of organic matter in surface water to groundwater have shown higher hydrogen to carbon ratios that indicate the groundwater contains more labile forms of organic carbon (McDonough et al., 2022). Sources of labile carbon to karstic groundwater remain poorly characterized; however, recent global estimates of dark primary production in carbonate aquifer ecosystems

(0.11 Pg C y$^{-1}$; Overholt et al., 2022) suggest chemoautotrophy may have a larger contribution than previously thought.

The Upper Floridan Aquifer (UFA) is one of the largest and most hydrologically productive karstic aquifers in the world, with an area of ~260,000 km$^2$ and depth of up to 500 m below the surface (Miller, 1986, 1997; Williams and Kuniansky, 2016). Most of the UFA is confined by the Hawthorn Group, which consists of siliciclastic sands and clays interbedded with

thin carbonate units (Scott, 1988). At the erosional edge of the Hawthorn Group in northwest Florida, the confining unit has been removed by erosion, promoting extensive surface water–groundwater exchange. The presence of numerous hydrological springs throughout this region provides direct access to groundwater that has had varying degrees of interaction with surface water and an experimentally tractable system to investigate the effects of hydrogeochemistry on microbial biomass and community metabolism. In this study, we tested the hypothesis that groundwater residence time and the

availability and quality of organic matter correlate with microbial biomass, metabolism, and production. This concept was examined by studying ten locations (Fig. 1) that discharge groundwater with differing organic matter characteristics and estimated residence times ranging from ~18 h to ~40 years (Martin and Dean, 1999; Martin et al., 2016). The effects of mixing with surface water bodies were studied by sampling water from springs that reverse flow under flood stage conditions (Gulley et al., 2011) as well as the Santa Fe River before and after it emerges from the subsurface after ~6 km

(Moore et al., 2009) of conduit flow and interaction with groundwater. Our results from the UFA microbial communities



furnish new insight on carbon flow and partitioning over an energy gradient ranging from nutrient-rich surface waters to oligotrophic groundwaters that had been stored in the aquifer for decades.

## 2 Materials and methods

### 2.1 Site description

Water from 10 springs and a river sink-rise system (O'Leno State Park) in North Central Florida (Fig. 1) were sampled between December 2018 and December 2022 (Table S1). Based on the geochemical properties of the groundwater discharged from the springs and extent of interaction with surface waters, the sites investigated can be classified into three groups (Flint et al., 2021): (1) the Ichetucknee springs group, which includes Blue Hole Spring (BH), Head Spring (HS), Mission Spring (MS), Coffee Spring (CS) and Devil´s Eye Spring (DE); (2) the reversing springs group, which includes

Madison Blue Spring (MB), Peacock Springs (PKS), Little River Spring (LRS), and Gilchrist Blue Springs (GBS); and (3) the Santa Fe River sink-rise system, where the River Sink (RS) and the River Rise (RR) are located (Fig. 1). Blue Hole Spring, Madison Blue Spring and River Rise are first magnitude springs (discharge $> 2.8$ m$^3$ s$^{-1}$); Gilchrist Blue Springs (vents 1 and 2), Little River Spring, Head Spring, and Devil's Eye Spring are second magnitude (discharge 2.8-0.28 m$^3$ s$^{-1}$); and Peacock Springs, Mission Spring, and Coffee Spring are third magnitude (discharge $< 0.28$ m$^3$ s$^{-1}$).


Springs of the Ichetucknee springs group (blue symbols in Fig. 1) show low temporal geochemical variability, are located far from the main recharge area, and have a low degree of mixing with surface waters (Martin et al., 2016; Henson et al., 2017). The Ichetucknee springs group is comprised of two subgroups, with groundwater from springs in subgroup I (Head Spring, Blue Hole Spring, and Coffee Spring) having shorter apparent ages (~30 years) and higher average dissolved oxygen (DO;

~3 mg L$^{-1}$) in comparison to springs in subgroup II (Mission Spring and Devil´s Eye Spring), which have apparent ages of ~40 years and average DO concentrations of ~0.5 mg L$^{-1}$ (Martin and Gordon, 2000). The Ichetucknee springs group discharge water draining a 960 km$^2$ springshed (Katz et al., 2009), approximately half of this area is located within the unconfined portion of the UFA, and the watershed is dominated by diffusive recharge with minimal point recharge. Approximately half of the landscape in the Ichetucknee springshed is forested and about one quarter is used for agriculture

(Katz et al., 2009).

When water levels of the spring vents in the reversing springs group (Peacock Springs, Madison Blue Spring, Little River Spring, and Gilchrist Blue Springs; green symbols in Fig. 1) are higher than that of the receiving water, these springs discharge clear, fresh groundwater to spring runs. However, increases in the river stage during periods of high discharge may

raise surface water elevations above groundwater heads, causing a reversal of spring flow direction that injects DO- and dissolved organic carbon- (DOC) rich surficial waters into subsurface conduits hydraulically connected to the spring vent (Gulley et al., 2011). At Madison Blue Spring, the river water intrusions have been documented at distances of at least 1 km



from the vent (Brown et al., 2014). Depending on rainfall and the spring's physiographic characteristics, the frequency of reversals for a given spring can vary from a few times a year to once every few years, and the effects of reversals on groundwater chemistry have been shown to persist for a period of at least 1 month (Brown et al., 2014, 2019).

In the Santa Fe Sink and Rise system (orange symbols in Fig. 1), the Santa Fe River flows over a confined portion of the UFA to a region that is unconfined; this boundary is a topographic feature known as the Cody Scarp (Puri and Vernon, 1964). At the River Sink, river water enters a sinkhole, flows underground for approximately 6 km through anastomosing water filled caves (herein called conduits) that have been extensively mapped by cave divers, and reemerges at the River Rise (Fig. 1) (Moore et al., 2009). Water transit times through the conduit system from the River Sink to the River Rise are estimated to be from 18 h to six days, depending on river stage (Martin and Dean, 1999). At lower rates of river discharge (i.e., $< {\sim}15$ m$^3$ s$^{-1}$ at River Rise), there is a net gain of water in the conduits from groundwater that flows from the rock matrix porosity (Martin and Dean, 2001; Flint et al., 2023). When discharge is $> {\sim}15$ m$^3$ s$^{-1}$ at River Rise, river water in the conduits flows into the matrix to recharge the aquifer. For simplicity, we subsequently use the descriptors 'low' and 'high' flow to distinguish between samples collected during these contrasting hydrological conditions at River Rise.







**Figure 1.** Locator map of the sites sampled in this study. MB: Madison Blue Spring; PKS: Peacock Spring; LRS: Little River Spring; DE: Devil's Eye Spring; HS: Head Spring; BH: Blue Hole Spring; MS: Mission Spring; CS: Coffee Spring. GBS: Gilchrist Blue Springs; RS: Santa Fe River Sink; RR: Santa Fe River Rise. The figure was generated using QGIS with base map data from © OpenStreetMap contributors 2021. Distributed under the Open Data Commons Open Database License (ODbL) v1.0.

**2.2 Groundwater sampling**

Spring discharge was sampled by performing hydrocasts from a canoe with a 5 L Niskin bottle (General Oceanics Inc., Miami, FL) or by pumping with a peristaltic pump (GeoTech) through PVC tubing led from the spring vent to the shore. Prior to deployment, the Niskin bottle was thoroughly cleaned with 10% (v/v) bleach followed by thorough rinsing with autoclaved deionized (DI) water. The PVC tubing was decontaminated by circulating a solution of 5% (v/v) hydrogen



peroxide for at least 1 minute and then rinsed with DI. Water samples collected in the Niskin bottle were carefully transferred to sterile 100 ml BOD bottles (radioisotope incorporation analyses) or 40 ml serum bottles (DIC production and oxygen consumption analyses) via clean silicon tubing that was inserted into the bottles. Each bottle was filled with sample

until no headspace remained and sealed with a glass or butyl stopper. To determine the initial DIC concentration, the water sample was filtered using a 0.1 µm PVDF syringe filter (Merck Milllipore, Cork, Ireland) and collected in 20 ml glass DIC bottles with no headspace. For transport to the laboratory, the samples were stored in an insulated cooler that contained water from each location to minimize temperature changes prior to analysis, which was always within 4 h of collection.

The POC and $\delta^{13}$C isotopic composition of POC ($\delta^{13}C_{POC}$) were analyzed using samples collected on 25 mm glass fiber filters (GF/F; Whatman). All sampling equipment was thoroughly cleaned with detergent and soaked overnight in 15 % HCl, followed by rinsing six times with Milli-Q water. Residual organic carbon on the tweezers and glass fiber filters was removed by combustion at 450 ºC for 4 h in a muffle furnace. Particles were purged from the tubing by pumping spring water through it for several minutes before attaching the filter housing and collecting the particulate samples. After filtration,

the samples were kept chilled upon return to the laboratory and stored at -20ºC until processed.

## 2.3 Water properties and organic carbon content

Water temperature, pH, ORP, specific conductivity, dissolved oxygen, turbidity, and depth data were obtained with a ProDSS multiparameter meter (YSI Inc.) that was calibrated prior to deployment. The data presented were collected at depths that were in the immediate vicinity and as close as possible to the vent discharge point.


DOC concentrations were measured on a Shimadzu TOC-V CSN total carbon analyzer. Three-dimensional fluorescence spectra were obtained using a Hitachi F-7000 Fluorescence Spectrophotometer across an excitation and emission range from 240–450 nm (5 nm intervals) and 250–550 nm (2 nm intervals), respectively. Three variables were parameterized from the spectral matrix data to describe organic matter quality: HIX (Humification index), BIX (Biological index) and FI

(Fluorescence index). HIX differentiates between organic matter with humic characteristics (> 16) compared with less humic characteristics approximating autochthonous aquatic bacterial origin (< 4); BIX differentiates dissolved organic matter of allochthonous (≤ 0.6) and autochthonous (~0.8–1.0) origin; and FI distinguishes fulvics of terrestrial origin (≤ 1.4) from those produced by microbes (≥ 1.9) (Flint et al., 2023).

Filters for POC, $\delta^{13}C_{POC}$, and the procedural blanks were decarbonated for 3 h under HCl fumes, dried at 40 ºC, and stored at -20 ºC until analyzed. Measurements for POC and the $\delta^{13}C_{POC}$ were made using a Thermo Electron DeltaV Advantage isotope ratio mass spectrometer coupled with a ConFlo II interface linked to a Carlo Erba NA 1500 CNHS Elemental Analyzer. All carbon isotopic data are expressed in standard delta notation relative to Vienna Peedee Belemnite (VPDB).



## 2.4 Cell and biomass concentration

Estimates of cell concentration were performed with 15 ml water samples that were preserved with formalin (final concentration 4 % v/v) immediately after collection. The fixed samples were subsequently vacuum filtered (< 23.7 kPa) through black 25 mm 0.2 µm pore size Isopore polycarbonate filter (Millipore) and a 0.45 µm pore size nitrocellulose backing filter (Whatman). DNA-containing cells on the filters were stained for 15 minutes with a 25× SYBR Gold stain (Invitrogen) solution that was diluted in 0.2 µm-filtered TBE (Tris Borate EDTA). After staining, the filter towers were

washed with clean TBE, vacuum was applied to filter the excess material, and the filters were mounted on glass microscope slides with a drop of a 1:1 antifade solution (0.1 % w/v phenylenediamine:glycerol). The stained cells were visualized and digitized using a Nikon Eclipse Ni-E epifluorescence microscope equipped with a C-FL GFP filter set (excitation 450–490 nm; dichroic mirror 495 nm; emission 500–550 nm) and 4.2 megapixel Zyla 4.2 PLUS camera.

For each sample, digital images were obtained for 40 random fields of view. To maximize depth of field for the analysis, a z-stack of 40 images was captured in 0.4 µm intervals (z-range of 16 µm) and compiled into a single image file for each field of view. The number, size, and shape of epifluorescent cells in the acquired images were determined by software-assisted tracing and individual observations were measured using NIS-Elements AR v4.51.00 (Nikon Inc.). The number of cells were determined based on the "brightest pixel detection" algorithm. For shape and size, a uniform "threshold cut-off" value was

not applied to all samples due to differences in background fluorescence among the samples. Instead, light threshold and contrast were manually adjusted for each sample in tandem with histogram analysis of binned intensity data. This allowed visual verification that the threshold parameters selected were processing data collected from individually stained cells. Contiguous pixels with intensity values above the threshold cut-off were used to define a particle boundary, from which the area, circularity, equivalent diameter, major axis length, and minor axis length were calculated by the software. Spherical

volume based on the major axis length was used to estimate biovolume following the assumption of equivalent diameter. Estimates of cell carbon using biovolume data were calculated according to Verity et al., 1992 and their conversion of 0.36 pg C µm$^{-3}$ for 10 µm$^3$ cells.

The 50 ml water samples for measurement of cellular ATP were collected in triplicate and sequentially filtered through 0.2

µm pore size Millex MF Millipore MCE membranes and 0.1 µm pore size Millex Durapore PVDF membrane filters using 60 ml syringes. Immediately upon return to the laboratory, the filters were stored at -20 ℃ and processed less than 48 h after collection. ATP was extracted from cells on the filters using the BioTherma ATP Biomass Kit HS and following the manufacturer´s instructions except for the following modification: 500 µl of Extractant B/S was added to and used to extract ATP from each filter by purging the material into a test tube with an air-filled sterile syringe. Subsequently, 100 µl of the

extracted material was mixed with 400 µl of reconstituted ATP reagent HS and immediately measured using a Turner BioSystems E5331 luminometer. The data presented for each filtered sample are the average values from three technical



replications. The amount of ATP in each sample was calculated as follows: $ATP_{smpl} = I_{smp} / (I_{smp} + std - I_{smpl})$, where $ATP_{smpl}$ is the amount of ATP in the sample (in pmol), $I_{smp}$ is the sample tube intensity in relative luminosity units (RLUs), and $I_{smp} + std$ is the intensity (in RLUs) in the sample tube after adding 10 µl of the $10^{-7}$ mol $L^{-1}$ internal ATP standard. ATP

concentration was converted into carbon biomass based on a molar ratio 250:1 for C:ATP (Karl, 1980).

**2.5 Microbial respiration**

Rates of oxygen consumption and production of dissolved inorganic carbon (DIC) were determined at in situ temperatures and calculated from the slope of linear regression models for concentration data obtained during time course experiments.

Oxygen concentration was measured in triplicate using stoppered 40 ml serum bottles containing OXSP5 oxygen sensor spots (Pyroscience) and no headspace. Killed controls were prepared in triplicate by amending samples with benzalkonium chloride to a final concentration of 0.01% (w/v). Bottles were incubated in an Innova 44 incubator (New Brunswick) at 21° C, except for samples collected at River Sink during low flow, which were incubated at the in situ temperature of 15° C. At daily intervals, oxygen concentration was measured using a calibrated FireSting®-O2 (1 channel) fiber-optical oxygen meter

(Pyroscience). A serum bottle containing deionized water was incubated with the samples to serve as the temperature reference for the oxygen measurements.

Upon return to the laboratory, the sample bottles for measurement of DIC production were incubated a 21 ℃ (except River Sink during low flow, which was incubated at 15 ℃). After 24, 48, and 96 h of incubation, triplicate bottles of the water

were filtered through a 0.1 µm pore size syringe filter (Merck Milllipore, Cork, IRL) into clean 20 ml DIC vials. Vials were stored at 4 ℃ and measured within 2 weeks of terminating the experiment. Total DIC was measured using a UIC (Coulometrics) 5017 $CO_2$ coulometer coupled with an AutoMate automated carbonate preparation device (AutoMateFX, Inc). Approximately 5 ml of sample was weighed into septum top tubes and placed into the AutoMate carousel. Acid and $CO_2$-free nitrogen carrier gas was then injected into the sample vial through a double needle assembly and evolved $CO_2$ is

carried through a silver nitrate scrubber to the coulometer where total carbon is measured.

The respiratory quotient (RQ) was determined by dividing the molar rate of DIC production by the rate of $O_2$ consumption. The average oxygen utilization rate (OUR) was calculated for paired samples at the Sink Rise system by subtracting the oxygen concentration at River Sink from the oxygen concentration at River Rise and dividing by the estimated residence

time based on the hydrological stage (18 h for high flow and 6 days for low flow; Martin and Dean, 1999). For Head Spring and Devil's Eye Spring, OUR was calculated by assuming the initial oxygen concentration when recharge occurred to be atmospherically equilibrated water at 21° C and sea level (8.5 mg $L^{-1}$).



## 2.6 ³H-leucine and -thymidine incorporation

Radioassays consisted of 0.8 ml water samples that were placed into 1.5 ml microcentrifuge tubes and amended with 0.2 ml

of a solution containing ³H-thymidine (thymidine [Methyl-³H], 50.8 Ci mmol[-1] in sterile water) or ³H-leucine (L-leucine [4, 5-³H], 160 Ci mmol[-1] in ethanol water 2:98; Perkin-Elmer). The final concentration for each was 20 nM, which corresponded to 1 µCi per sample for the ³H-thymidine assays and 3.2 µCi per sample for ³H-leucine. Killed controls were prepared by adding 200 µl of 50 % (v/v) formalin to designated water samples. Six replicates per time point were established for the experimental and control groups. All samples were incubated at 21° C in the dark.


At designated time intervals during the experiment (48 h for Devil's Eye, Madison Blue and the River Rise at high flow; 24 h for all other samples) subsets of the samples were killed by adding 200 µl of 50 % (v/v) formalin and storing at 4° C. Acid insoluble macromolecules were precipitated by adding 200 µl of an ice-cold solution of 100 % (w/v) TCA followed by centrifugation at 15000 $x\ g$ for 15 min. Samples were then sequentially washed with 5 % (w/v) TCA and 70 % (v/v) ethanol,

with a 5 min centrifugation at 15000 $x\ g$ after each wash. The washed pellets were suspended in 1 ml of scintillation cocktail (CytoScint, MP Biomedicals), the tubes were placed into scintillation vials, and radioactivity was measured on the ³H channel of a Beckman LS6500 scintillation counter for 10 min. To determine disintegrations per minute from the count per minute data, the counting efficiency was calculated using a quench curve and ³H-Toluene standard (Perkin-Elmer) with a specific activity of 2.552 dpm g[-1] (20 µl). The curve was based on data generated by mixing 20 ml of scintillation cocktail

(CytoScint) with 42,335 dpm of ³H-Toluene (MP Biomedicals), and 0 %, 0.25 %, 0.5 %, 1 %, 1.5 %, 2 %, 2.5 %, 4 % and 5 % acetone (v/v).

To convert ³H-thymidine and ³H-leucine incorporation rates to cell carbon and estimate heterotrophic production, standard conversion factors of 2.0 x $10^{18}$ cells mol[-1] were used for ³H-thymidine (Fuhrman and Azam, 1980) and 1.42 x $10^{17}$ cells mol[-1]

for ³H-leucine (Chin-Leo and Kirchman, 1988). Cellular carbon content was based on values estimated from biovolume. Bacterial growth efficiency (BGE) was calculated as the quotient between carbon incorporated into biomass and the sum of carbon incorporated into biomass plus that respired as DIC.

## 3 Results

### 3.1 Hydrogeochemistry

The groundwater discharged from sites in the Ichetucknee springs group (Head Spring, Blue Hole Spring, Coffee Spring, Devil's Eye Spring, and Mission Spring) had little interaction with surface waters during its subsurface residence time and these springs discharge continuously at rates that vary by less than a factor of three (Martin and Gordon, 2000; Katz, 2004; Kurtz et al., 2015). In comparison, members of the reversing springs group (Madison Blue Spring, Gilchrist Blue Spring,



Peacock Springs, and Little River Spring) experience reversal of flow when river water levels exceed groundwater heads.

For instance, there were eight occasions from 2018 to 2022 when the stage of the Withlacoochee River resulted in flow reversal of Madison Blue Spring. We had opportunities to sample Madison Blue Spring and Peacock Springs during periods when they were reversing (Table 1 and Table S1) as shown by the USGS gauging station (02319302) which show reversals as negative discharge values and discharging flow as positive values. Observed DO concentrations during reversals (7.21 and 3.86 mg L$^{-1}$, respectively) were at least twice as high as the values when discharge was positive. The samples we analysed

from Madison Blue Spring for POC, DIC, oxygen consumption, and heterotrophic production were all collected in 2022 and at timeframes of 1 to 104 d after it had transitioned from negative to positive discharge.

The physical and chemical properties of groundwater discharged from the Ichetucknee and reversing springs groups showed minimal variation over 48 months of observation, and their clear (turbidity values near 0 FNU), circumneutral waters had an

average temperature of 21.83 ± 0.07° C (n=65; ± the standard error; Table 1). There are statistically significant differences in specific conductance (SpC; ANOVA, $p < 0.05$), oxidation-reduction potential (ORP; $p < 0.001$), and DO concentration ($p < 0.001$) among the Ichetucknee and reversing springs groups. Discharge from Peacock Springs and Little River Spring had the highest SpC values, and Devil's Eye Spring and Mission Spring had the lowest DO concentration (average of 0.21 and 0.55 mg L$^{-1}$, respectively; Table 1). Water quality parameters from the Santa Fe Sink and Rise system ranged widely

depending on hydrological conditions (Table 1). When groundwater was transported into the conduit during periods of low flow, water at River Rise had characteristics similar to regional groundwater. For example, temperature that tended to approach 21° C and SpC higher than Santa Fe River source waters. Variation in these physical water properties during periods of high flow coincided with increasing DOC concentration and HIX (Table 2), indicating that the distinct hydrological regimes in the river sink-rise system were associated with appreciable changes in the quantity and quality of

organic matter.




| Parameter | Ichetucknee Subgroup I | | | Ichetucknee Subgroup II | | Reversing Springs Group | | | | | | | Santa Fe System | |
|---|---|---|---|---|---|---|---|---|---|---|---|---|---|---|
| Spring designation | HS (n=11) | BH (n=4) | CS (n=4) | DE (n=14) | MS (n=6) | GBS 1 (n=5) | GBS 2 (n=3) | PKS (n=6) | PKS (During reversal) (n=1) | LRS (n=4) | MB (n=8) | MB (During reversal) (n=1) | RS (n=9) | RR (n=13) |
| Water temperature (ºC) | 21.71 ± 0.01 | 21.60 ± 0 | 21.85 ± 0.04 | 21.78 ± 0.01 | 21.70 ± 0 | 22.56 ± 0.03 | 22.50 ± 0.00 | 21.63 ± 0.03 | 25.7 | 21.93 ± 0.09 | 20.90 ± 0.02 | 15.70 | 14.7–27.10 | 15.7–26.30 |
| pH | 7.26 ± 0.04 | 7.42 ± 0.07 | 7.40 ± 0.07 | 7.28 ± 0.04 | 7.25 ± 0.07 | 7.24 ± 0.04 | 7.37 (n = 1) | 7.35 ± 0.10 | 6.48 | 7.14 ± 0.06 | 7.46 ± 0.08 | 6.26 | 6.09–7.79 | 6.14–8.25 |
| ORP (mV) | 161 ± 6 | 169 ± 14 | 178 ± 7 | 143 ± 6 | 133 ± 8 | 189.95 ± 9.55 | 182.5 (n = 2) | 116 ± 10 | 206 | 133 ± 7 | 210 ± 3 | 190 | 103–207 | 30.4–205 |
| DO (mg L⁻¹) | 3.70 ± 0.01 | 1.46 ± 0.04 | 2.75 ± 0.14 | 0.21 ± 0.02 | 0.55 ± 0.04 | 4.60 ± 0.04 | 4.55 ± 0.02 | 1.84 ± 0.3 | 3.86 | 1.42 ± 0.06 | 1.62 ± 0.11 | 7.21 | 4.02–6.93 | 0.01–6.44 |
| Sp. Cond. (µS cm⁻¹) | 346.0 ± 1.5 | 309.8 ± 1.9 | 301.0 ± 1.8 | 347.4 ± 1.2 | 323.5 ± 2.1 | 385.4 ± 2.6 | 390.6 ± 8.48 | 430.4 ± 2.8 | 64.3 | 417.1 ± 6.2 | 299.0 ± 2.7 | 76.9 | 71.8–319 | 78.2–532.3 |
| Salinity (PSU) | 0.166 ± 0.001 | 0.148 ± 0.001 | 0.142 ± 0.001 | 0.166 ± 0.001 | 0.153 ± 0.001 | 0.184 ± 0.001 | 0.187 ± 0.004 | 0.207 ± 0.002 | 0.03 | 0.200 ± 0.011 | 0.145 ± 0.001 | 0.040 | 0.030 – 0.150 | 0.040 – 0.260 |
| TDS (mg L⁻¹) | 224 ± 1 | 201 ± 1 | 196 ± 1 | 226 ± 1 | 210 ± 1 | 250 ± 2 | 254 ± 5 | 280 ± 2 | 42 | 271 ± 4 | 195 ± 2 | 50 | 47–208 | 51–346 |
| Turbidity (FNU) | 0 | 0.25 ± 0.12 | 0.05 ± 0.03 | 0.31 ± 0.22 | 0.16 ± 0.10 | 0.07 ± 0.04 | 0 | 0 | 3.15 | 0.29 ± 0.10 | 0.28 ± 0.12 | 10.90 | 0.77–26.37 | 0.32–9.47 |
| DIC (µg C g⁻¹) | 40.76 ± 0.05 | N.d. | N.d. | 39.3 ± 0.1 | N.d | N.d. | N.d | N.d | N.d. | N.d. | 30.51 ± 0.03 | N.d. | 28.39 – 2.27 | 33.64 – 10.21 |
| Depth to vent from surface (m) | 4.90 | 11.42 | 0.71 | 3.56 | 1.29 | 5.47 | 5.22 | 6.01 | 8.53 | 3.92 | 7.2 | 13.26 | 1.30 | 11.13 |
| Location (Lat, Long, WSG 84) | 29.984° N 82.762° W | 29.981° N 82.759° W | 29.959° N 82.775° W | 29.974° N 82.760° W | 29.976° N 82.758° W | 29.830° N 82.683° W | 29.830° N 82.681° W | 30.122° N 83.133° W | 30.122° N 83.133° W | 29.997° N 82.966° W | 30.481° N 83.244° W | 30.481° N 83.244° W | 29.912° N 82.573° W | 28.874° N 82.591° W |

**Table 1.** Summary of basic physical and geochemical data collected from each location sampled from 2018 to 2022. ORP: Redox potential; DO: Dissolved oxygen; Sp. Cond: Specific conductivity; TDS: Total dissolved solids; and N.d.: No data

Results of a one-way ANOVA and post-hoc Tukey HSD analysis showed POC concentration was significantly higher at the River Sink and River Rise in comparison to the other sites sampled (ANOVA; $p < 0.001$, F = 22.007; Fig. 2 a). Head Spring, Devil's Eye Spring, and Madison Blue Spring had the lowest concentrations of POC (< 0.03 mg C L⁻¹; ANOVA, $p < 0.001$,



F = 22.007) and DOC (0.2 to 0.4 mg C L$^{-1}$; Table 2). The heaviest $\delta^{13}$C value for POC was observed at Madison Blue Springs (average of -28.7 ± 0.3 ‰), which was enriched by approximately 2 ‰ and 4 ‰ relative to that for Devil's Eye

Spring (-31.6 ± 0.3 ‰) and Head Spring (-32.5 ± 1.4 ‰), respectively (Fig. 2 b). Though values for $\delta^{13}$C$_{POC}$ were isotopically lighter in samples from the Santa Fe Sink and Rise system during low flow, the differences are not statistically different from those at high flow (Fig. 2 b). Dissolved organic matter with HIX values below 10, BIX above 0.8, and FI above 1.9 is generally assumed to be of high quality (e.g. Flint et al., 2023). HIX values indicate that the Ichetucknee springs group had the lowest humic content, and higher values for FI and BIX imply the organic matter was of higher quality relative

to the other springs sampled (Table 2). Though much higher concentrations of DOC were observed in water from the river sink-rise system (Flint et al., 2023), the HIX, FI, and BIX values are indicative of low quality organic matter. Higher concentrations of DOC and POC were observed during periods of high flow at River Sink, but there was not a statistical difference in POC concentration with flow stage at River Rise (Fig. 2 a).





| | RS | RS-H | RR | RR-H | MB* | HS | DE |
|---|---|---|---|---|---|---|---|
| Subsurface residence time | N.a. | N.a. | ~6 d[e] | ~18 h[e] | N.d. | ~30 y[f] | ~40 y[f] |
| DOC (mg C L⁻¹)[a] | 3.9 – 30.1 | 24.6 – 58.8 | 2.4 – 23.5 | 21.5 – 50.2 | 0.3 – 14.4 | 0.2 – 0.4 | <0.1 – 1.0 |
| HIX[b] | 8.28 – 24.46 | 16.06 – 30.45 | 13.42 – 23.77 | 17.40 – 30.40 | 5.07 – 20.21 | 1.90 – 4.40 | 2.52 – 13.61 |
| BIX[b] | 0.42 – 0.53 | 0.40– 0.45 | 0.46– 0.54 | 0.40– 0.47 | 0.47 – 0.76 | 0.77 – 0.94 | 0.55 – 0.75 |
| FI[b] | 1.37– 1.51 | 1.34– 1.43 | 1.44– 1.51 | 1.35– 1.43 | 1.44 – 1.71 | 1.84 – 2.04 | 1.54 – 1.81 |
| Biovolume cell carbon (fg C cell⁻¹) | 36.3 ± 0.1 | N.d. | 50.1 ± 0.2 | N.d. | 105 ± 4.9 | 64.9 ± 0.7 | 52.2 ± 0.9 |
| OUR (mg L⁻¹ d⁻¹) | N.a | N.a | 0.5517 | 6.4133 | N.d. | 0.0004 | 0.0006 |
| O$_2$ consumption (mg L⁻¹ d⁻¹)[c] | 0.18 ± 0.02 | 0.32 ± 0.01<br>0.319± 0.006 | 0.137 ± 0.008 | 0.38 ± 0.02 | 0.17 ± 0.02<br>0.10 ± 0.01<br>N.d.<br>0.19 ± 0.03 | 0.19 ± 0.01 | 0.10 ± 0.01 |
| DIC production (mg C L⁻¹ d⁻¹)[c] | 0.36 ± 0.07 | 0.096 ± 0.009<br>0.08 ± 0.01 | 0.49 ± 0.05 | 0.16 ± 0.03 | N.d.<br>0.09 ± 0.01<br>N.d.<br>0.19 ± 0.03 | BLD | BLD |
| Respiratory Quotient | 2.66 | 0.39<br>0.35 | 4.85 | 0.56 | N.d.<br>1.23<br>N.d.<br>0.62 | N.d. | N.d. |
| Heterotrophic production (µg C L⁻¹ d⁻¹)[d] | 4653 ± 88 | 18328 ± 110 | 1284 ± 85 | 14225 ± 343 | 60 ± 6<br>N.d.<br>42 ± 2<br>90 ± 11 | 576 ± 89 | 198 ± 22 |
| Cell-specific heterotrophic production (pmol C cell⁻¹ h⁻¹)[d] | 0.1242 | 0.1046 | 0.0135 | 0.1135 | 0.0056<br>N.d.<br>0.0071<br>0.0262 | 0.2607 | 0.0485 |
| Specific growth rate (d⁻¹)[d] | 0.99 | 0.83 | 0.08 | 0.65 | 0.02<br>N.d.<br>0.02<br>0.07 | 1.16 | 0.27 |
| Doubling time[d] (h) | 17 | 20 | 215 | 25 | 1028<br>N.d.<br>856<br>232 | 14 | 62 |
| BGE[d] (%) | 92.9 | 99.5 | 72 | 98.9 | N.d.<br>N.d.<br>39.3<br>32.1 | 87[g] | 81[g] |
| Leu:TdR | 3.10 | 3.56 | 3.86 | 4.06 | 0.42 | 6.24 | 2.72 |



| | | | | N.d.<br>0.29<br>0.44 | | |
|---|---|---|---|---|---|---|

**Table 2.** DOC concentration and quality, biogeochemical data, and microbial physiological properties derived from the rate data. RS-H and RR-H: River Sink and Rise under high flow conditions, respectively. [a] Data from Flint et al., 2021; [b] DOC quality data were collected between August 2018 through March 2022. [c] Error (±) is based on the slope uncertainty and not the standard error; [d] Based on $^3$H-leucine incorporation and DIC production rates; [g] Based on the oxygen consumption data and a theoretical RQ = 1.2 (Berggren et al. 2012); Residence time data are from [e] Martin and Dean, 1999 and [f] Martin et al., 2016. Multiple values for a given parameter represent
independent measurements from different dates in chronological order (see Table S1), and for Madison Blue Spring, these samples correspond to 51, 1, 104, and 91 d, respectively, after a reversal. N.d.: no data. N.a.: Not applicable; BLD: Below the level of detection of this analytical procedure.

**Figure 2.** POC concentration (a) and the $\delta^{13}$C of POC (b) in samples from the springs and sink-rise system. Lowercase letters indicate the
significance groups based on a Tukey HSD test. RR-H and RS-H are samples taken during high flow at the sink-rise system.



## 3.2 Microbial Cell and Biomass Concentrations

The results of a one-way ANOVA ($p < 0.001$, $F = 28.35$) indicate significant differences in cell concentration means among samples from the ten springs and river sink-rise system analyzed (Fig. 3 a). The lowest cell concentrations were observed in samples from the Ichetucknee springs group, which ranged from $4.83 \pm 0.50 \times 10^6$ cells $L^{-1}$ at Blue Hole Spring to $1.29 \pm$
$0.05 \times 10^7$ cells $L^{-1}$ at Devil's Eye Spring. Springs that had periodically experienced flow reversal (i.e., Madison Blue Spring, Peacock Springs, and Little River Spring) generally had higher average cell concentrations at baseflow than those in the Ichetucknee springs group, but only samples from Little River Spring were significantly higher (Tukey HSD test, $p < 0.001$). Cell concentration means for the River Sink and River Rise ($6.08 \pm 0.8$ and $4.35 \pm 0.4 \times 10^8$ cells $L^{-1}$, respectively) are significantly higher ($p < 0.001$) than all other samples except those from Little River Spring and Madison Blue Spring during
reversed flow (Fig. 3 a).





**Figure 3.** Cell abundance and biomass estimates in samples collected from various springs and sink-rise system. (a) Log10-transformed data for cell abundance. Sample order in panels (a)–(d) is based on sites with increasing mean cell concentrations based on direct epifluorescent microscopic counts. (b) Microbial biomass based on cell volumetric data derived from microscopic observations. (c)



Microbial biomass based on the concentration of cellular ATP retained in the 0.2 µm fraction. (d) ATP concentration per cell (in zeptomol). Samples labeled with the suffix -Rev indicate those taken during a spring reversal and the suffix -H indicates those taken during high flow at River Rise. Lower case letters indicate the significance groups based on a Tukey HSD test.

From a total of 198,700 individual biovolume measurements, 1,029 extreme outliers [i.e., values of Q3 + (3 × IQR) or Q1 -
       (3 × IQR)] were identified among the samples and removed, and subsequent analyses were conducted on an average of
       $19,762 \pm 6,160$ observations per sample. Average biovolume in the groundwaters and surface waters sampled was $0.1259 \pm$
       $0.0003$ µm$^{-3}$, ranging from a high of $0.36 \pm 0.02$ µm$^{-3}$ (Madison Blue Spring) to a low of $0.1010 \pm 0.0003$ µm$^{-3}$ (River Sink).
       A Kruskal–Wallis test indicated there were significant differences in biovolume among the samples ($p < 0.001$). Therefore,
cell carbon estimates were calculated separately for each spring. A sample from River Sink at low flow had the lowest
       average cellular carbon content ($36.3 \pm 0.1$ fg C cell$^{-1}$; Fig. S1), which is significantly lower than values at other sites (post-
       Hoc Dunn test; $p < 0.001$). The highest cellular carbon content of $105 \pm 5$ fg C cell$^{-1}$ was observed at Madison Blue Spring
       (Table 2) from a sample collected 83 d after the spring had reversed from negative to positive discharge. Multiplying cell
       carbon by cell concentration provides a total estimate of microbial biomass (Fig. 3 b), which is strongly correlated with cell
abundance (Spearman r = 0.988) (Fig. 3 a). Microbial biomass in samples from the sink-rise system (River Sink, 22,065 ng
       C L$^{-1}$; River Rise, 21,799 ng C L$^{-1}$; Fig. 3 b) was ~100-times higher than the Ichetucknee springs group (236 to 673 ng C L$^{-1}$
       $^{1}$). The quantity of carbon biomass in samples from Devil's Eye Spring and Head Spring corresponded to 3 % of the carbon
       in POC (Fig. 2 a), whereas much higher fractions of the POC were inferred to be microbial biomass in samples from
       Madison Blue Spring (24 %) and the sink-rise system (24 % and 11 % for River Sink and River Rise, respectively).


       Experiments that sequentially passed water samples through 0.2 and 0.1 µm pore size filters showed most extractable ATP
       (79 to 99 %) was associated with cells retained on the 0.2 µm pore sized filters. One exception was vent two of Gilchrist
       Blue Spring, where total extractable ATP in the 0.1 to 0.2 µm fraction (0.38 pM) exceed that in the > 0.2 µm fraction by ~2-
       fold (Fig. S2). Unexpectedly, cells in the > 0.2 µm fraction from Devil's Eye Spring contained ATP concentrations ~15
times higher than those observed in samples from other springs in the Ichetucknee springs group as well as reversing spring
       group, with values similar to those for the sink-rise system (Fig. S2). To exclude a technical error or contamination as
       explanations for the high ATP concentration measured at Devil's Eye Spring, repeat sampling conducted within one month
       of the initial observation confirmed that the cells contained higher cellular ATP concentrations than other springs. Dividing
       ATP by cell concentration provides an average estimate of ATP content per cell (Fig. 3 d), and these data show that cellular
ATP in samples from Devil's Eye Spring are significantly higher ($p < 0.001$; $1516 \pm 300$ zmol ATP cell$^{-1}$) than all other
       observations (range of 1.25 to 622.32 zmol ATP cell$^{-1}$). The trend in carbon biomass estimated from the cellular ATP
       concentration (Fig. 3 c) generally agrees with that based on biovolume (Fig. 3 b) but there are exceptions. For instance, low
       biomass is inferred for most Ichetucknee and reversing springs with positive rates of discharge, the latter of which
       significantly increased during periods of reversal (Fig. 3 c). However, biomass carbon for Devil's Eye Spring based on ATP
data was significantly higher than for the other Ichetucknee springs (Fig. 3 c) and not statistically different from River Rise





samples that contained ~15-fold more cells (Fig. 3 a). ATP-based biomass concentrations in samples from River Rise at low flow are significantly higher than values at high flow and ~3-fold lower than those observed for River Sink at low flow. As ATP is considered a proxy for viable biomass (e.g. Köster and Meyer-Reil, 2001; Oulahal-Lagsir et al., 2000), we used the ratio between biomass estimates derived from biovolume (Fig. 3 b) and ATP concentration (Fig. 3 c) to assess trends that

may be related to viability of the groundwater communities. ATP-based biomass in the reversing springs group ranged from 1.4 % (Madison Blue Spring) to 38.5 % (Gilchrist Blue Spring) of the biovolume-based biomass estimates, whereas values for the River Rise and River Sink were 4.6 % and 13.7 %, respectively. In contrast, ATP-based biomass was up to 4-fold higher than estimates inferred from biovolume for the Ichetucknee springs group (Blue Hole Spring, Coffee Spring, and Devil´s Eye Spring; 145 %, 163 % and 404 %, respectively; (Fig. 2 c).

**3.3 Microbial respiration**

The rate of oxygen consumption was highest in samples from River Rise and River Sink, with rates at high flow (0.382 and 0.322 mg L$^{-1}$ d$^{-1}$, respectively) being approximately twice those observed during low flow (0.137 and 0.178 mg L$^{-1}$ d$^{-1}$, respectively; Fig. 4 a; Table 2; Fig. S3). In contrast, rates of DIC production during low flow were at least three times higher than those observed during high flow conditions (Fig. 4; Table 2). The lowest oxygen consumption rates were in

groundwaters from Devil's Eye Spring (0.102 mg L$^{-1}$ d$^{-1}$) and samples taken at Madison Blue Spring 1 d after a period of flow reversal (31 March 2022; 0.101 mg L$^{-1}$ d$^{-1}$). Significantly higher rates of oxygen consumption (ANOVA F = 44.912, p < 0.001) were measured in discharge from Head Spring (0.189 mg L$^{-1}$ d$^{-1}$), as well as in samples from Madison Blue Spring (0.168 mg L$^{-1}$ d$^{-1}$, and 0.199 mg L$^{-1}$ d$^{-1}$; Fig. 4 a) that occurred 51, and 91 d, respectively, after it had transitioned from negative to positive rates of discharge. A higher rate of oxygen consumption in the 91 d sample from Madison Blue Spring

matched a DIC production rate that was ~2-fold higher than values observed immediately after the transition from reversing conditions (Fig. 4 b). Unfortunately, poor fit of the DIC data from Devil's Eye Spring and Head Spring to the regression models (r$^2$ ≤ 0.2; Fig. S4) coupled with no statistically significant change in concentration over time prevented an estimate of DIC production by this method. The observed molar ratio of DIC produced to oxygen consumed (i.e., respiratory quotient, RQ) for Madison Blue Spring 91 d after a reversal event (0.62) was approximately half that measured after one day of

positive discharge (1.23). At the sink-rise system, RQ was much higher during low flow (4.85 and 2.66 for River Rise and River Sink, respectively) than high flow (0.56 and 0.37, respectively; Table 2).

At low flow in the sink-rise system, the OUR (0.55 mg L$^{-1}$ d$^{-1}$) was higher but similar to measured rates of oxygen consumption (0.14 mg L$^{-1}$ d$^{-1}$); however, OUR at higher flow (6.41 mg L$^{-1}$ d$^{-1}$) was ~16-fold higher than measured oxygen

consumption rates (Table 2). The largest discrepancy documented was for members of the Ichetucknee springs group, which had measured rates of oxygen consumption (Fig. 4 a) three orders of magnitude higher than OUR (1.26 to 5.68 x 10$^{-4}$ mg L$^{-1}$





d$^{-1}$; Table 2).



**Figure 4.** Rates of oxygen consumption (a) and DIC production (b) in select springs and sink-rise system. Letters indicate significance groups based on a Tukey HSD test. DIC production data are not available for the Madison Blue Spring sample collected on 7 March 2022. Samples labeled with the suffix -H indicate those taken during high flow at River Sink and River Rise.

### 3.4 Heterotrophic Carbon Assimilation

Time series experiments that were performed to quantify the amount of $^3$H-leucine and $^3$H-thymidine incorporated into acid insoluble macromolecules provided data to estimate assimilatory metabolic and growth rates for heterotrophic members of the communities. Observed rates of $^3$H-leucine incorporation exceeded those for $^3$H-thymidine, except for the samples from Madison Blue Spring, where $^3$H-thymidine incorporation rates were up to 3-fold higher than those for $^3$H-leucine incorporation (Fig. S5). The highest rates of incorporation were observed in samples from the River Sink and River Rise that also contained the highest cell and biomass concentrations (Fig. 3). The molar ratio of $^3$H-leucine to $^3$H-thymidine



incorporation (Leu:TdR) was very low (< 0.45) for observations from Madison Blue Spring and much lower than the longest

residence time groundwater (Devil's Eye, 2.72; Table 2). The largest Leu:TdR ratios were observed in samples from the

River Sink (3.10), River Rise (4.06), and Head Springs (6.24).

To estimate doubling times from the $^3$H-leucine and $^3$H-thymidine incorporation data, we assumed that all cells in the

samples (Fig. 3 a) were viable and capable of incorporating the radiotracers into newly synthesized protein and DNA,

respectively. Therefore, the values derived correspond to maximum reproduction estimates for the cell populations.

Doubling times calculated from cell specific rates of $^3$H-leucine (Table 2) and $^3$H-thymidine incorporation negatively

correlate with dissolved oxygen (r = -0.79 to -0.86, p < 0.05). However, the separate radiotracers did not provide matching

reproduction rates and the values based on $^3$H-thymidine incorporation are 2 to 50-times shorter than those based on $^3$H-

leucine incorporation. Given that freshwater bacteria have shown preferential uptake of leucine over thymidine (e.g., Pérez et

al., 2010), we used the $^3$H-leucine incorporation rates to further evaluate rates of growth and cell carbon production in the

groundwaters and surface waters studied. Doubling times of 20 h at the River Sink were shorter than those at the River Rise,

and under low flow conditions, doubling time increased to nearly 9 days at River Rise (Table 2). Much longer doubling

times of 10 to 42 d were inferred at Madison Blue Spring (i.e., when discharge rates were positive) in comparison to

groundwater collected from the Ichetucknee springs group (14 h for Head Spring and 62 h for Devil's Eye Spring).


A standard leucine-to-carbon conversion factor was used to estimate heterotrophic productivity based on the rate of $^3$H-

leucine incorporation (Fig. 5). The values obtained for heterotrophic productivity on three dates at Madison Blue Spring

were the lowest observed (42.9 to 90.6 µg C L$^{-1}$ d$^{-1}$). Based on an ANOVA (p < 0.001, F = 144.183) and Tukey HSD post-

hoc analysis, heterotrophic productivity for Devil's Eye Spring (198 µg C L$^{-1}$ d$^{-1}$) and Head Spring (576 µg C L$^{-1}$ d$^{-1}$) were

significantly higher than at least two of the three observations from Madison Blue Spring (Fig. 5). The highest rates of

heterotrophic production were measured during high flow at the River Sink (18,328 µg C L$^{-1}$ d$^{-1}$) and River Rise (14,225 µg

C L$^{-1}$ d$^{-1}$), which were significantly different from (ANOVA p < 0.001, F = 188.553) and 4- and 11-fold higher, respectively,

than values during low flow (Fig. 5). Low flow conditions coincided with significantly higher rates of heterotrophic

production at the sink (ANOVA p < 0.001, F = 205.016; 4,653 µg C L$^{-1}$ d$^{-1}$) versus the rise (1,285 µg C L$^{-1}$ d$^{-1}$). Cell carbon

incorporation rates derived from $^3$H-thymidine follow a similar pattern among samples as those derived from $^3$H-leucine, but

in general, implied higher rates of heterotrophic production at most sites (Fig. S7). The trend for cell specific rates of

heterotrophic productivity based on the $^3$H-leucine data (Table 2) is quite different from that for bulk values (Fig. 5), and the

highest cell specific rates were observed at Head Spring, which exceeded those for surface waters at the River Sink by ~2-

fold.


Very high growth efficiencies were inferred from estimates of heterotrophic production ($^3$H-leucine data) and respiration

(DIC production data) for samples from the sink-rise system (93 to 99%), with a decreased BGE of 72% at River Rise under




low flow conditions. The lowest BGEs (39% and 32%) were found in samples collected from Madison Blue Spring at least three months after a period of flow reversal (Table 2). Although we were unable to empirically determine respiration rates

using the DIC concentration data collected from Devil's Eye Spring and Head Spring, BGE values calculated using the oxygen consumption rate and a theoretical RQ of 1.2 (Berggren et al., 2012) were high (81% and 87%, respectively) and implied that a lower proportion of the DOC utilized was lost as $CO_2$ in comparison to the groundwater sampled from Madison Blue Spring.

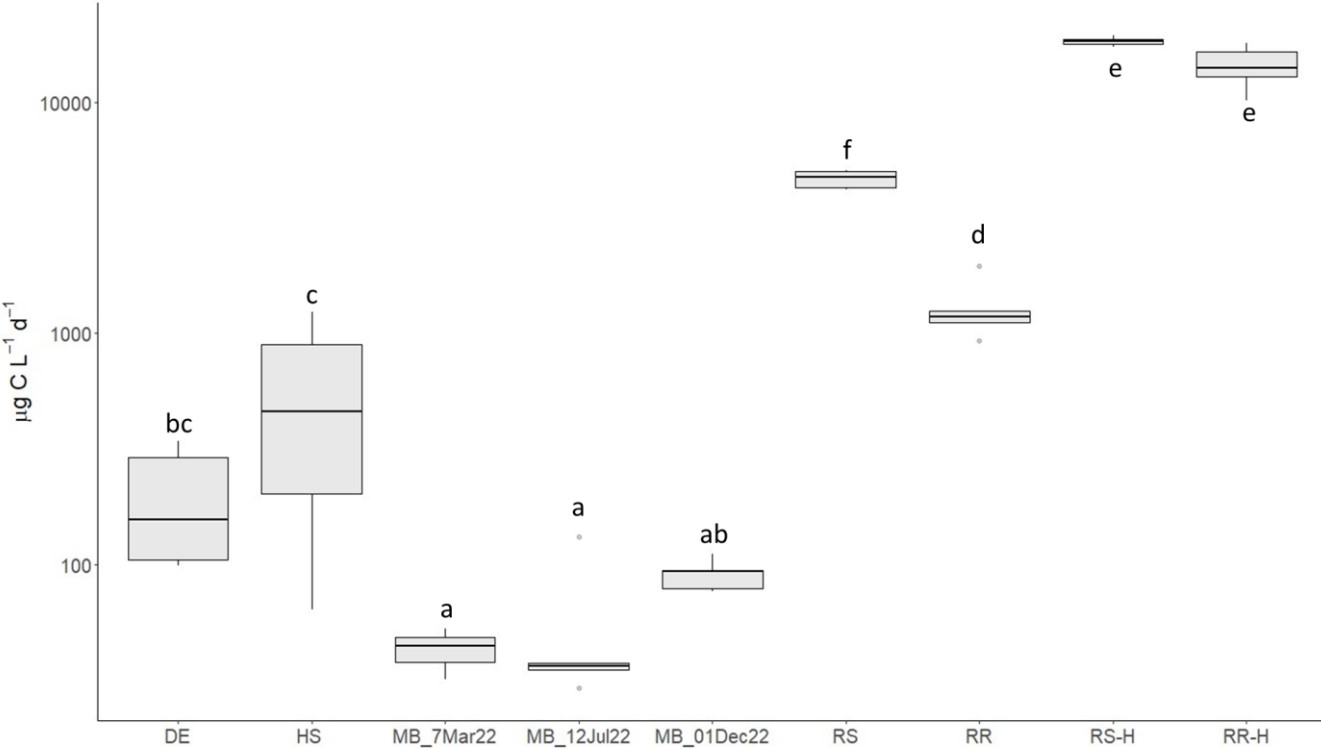

**Figure 5.** Rates of heterotrophic carbon production based on $^3$H-leucine incorporation data. Letters indicate significance groups based on a Tukey post-hoc analysis. Samples labeled with the suffix -H indicate those taken during high flow at River Sink and River Rise.

## 4 Discussion

### 4.1 Groundwater and surface water mixing in karst landscapes

Studies that have examined microbial production in aquifer ecosystems have deepened understanding of subsurface carbon

cycling and the biogeochemical processes contributing to groundwater quality (e.g., Wilhartitz et al., 2009; Hofmann and Griebler, 2018; Karwautz et al., 2022). Research over recent decades in silicate-dominated aquifer systems has contributed greatly to these discussions, but there has been less emphasis on landscapes composed predominantly of carbonate minerals (Covington et al., 2023). A key aspect of karstic aquifers are the fractures and conduits that enhance permeability and water





flow within their geological formations, facilitating relatively rapid exchange of dissolved gases, nutrients, organic matter,
and microbes, and between surface water bodies and groundwater. The groundwater discharged from springs in North
Central Florida tends to be organic carbon poor and suboxic in contrast to surface waters that are organic carbon-rich and
with DO near equilibration with atmospheric oxygen (Moore et al., 2009; Martin et al., 2016; Flint et al., 2021; Oberhelman
et al., 2023). Consequently, this region provides a model experimental setting to evaluate the effects of surface-derived
substrate delivery on microbial consumption of organic matter and biomass production in karstic groundwater.

**4.2 Abundance of microbes, biomass, and organic carbon**

Cell concentrations in groundwater from the Ichetucknee springs group (Fig. 3 a) are similar to those reported in previous
studies of springs in this region (Malki et al., 2020, 2021) and other oligotrophic karstic groundwaters ($\sim10^6$ to $10^7$ cells L$^{-1}$;
Farnleitner et al., 2005; Wilhartitz et al., 2007, 2009, 2013; Hershey et al., 2018). In comparison, water samples from the
reversing springs and sink-rise system had higher DO (Table 1), DOC (Table 2), cell ($\sim10^7$ to $10^9$ cells L$^{-1}$), and biomass
(Fig. 3 b and c) concentrations. Biomass estimates based on biovolume (Fig. 3 b) are largely congruent with those derived
using ATP data for most of the Ichetucknee springs and reversing springs when their discharge rates were positive (Fig. 3 c).
The ATP-based biomass data reveal an effect of surface waters on the microbial communities, showing significantly higher
values during periods of spring reversal and during hydrological conditions when groundwater mixing occurred at the River
Rise. For the Ichetucknee springs group, the ATP data provided higher biomass estimates in comparison to those derived
from biovolume, and the largest discrepancy is for Devil's Eye Spring, where the ATP-based biomass estimate is 4-times
higher (Fig. 3 b and c). Our data and related calculations imply that average cellular ATP contents in populations discharged
with suboxic, oligotrophic groundwater at Devil's Eye Spring (Fig. 3 d) were at least 4-fold higher than those at other sites
and approach those reported in laboratory cultured bacteria (e.g., 2,000 zmol; Thore et al., 1975). High ATP contents are
associated with cells that have large volumes and rapid metabolisms (e.g., Eydal and Pedersen, 2007), but neither of these
explanations are consistent with the biovolume or metabolic rate data (Table 2) observed at Devil's Eye Spring.

The quantity and quality (i.e., composition, chemical structure, and nutrient content) of organic matter is a crucial
determinant of heterotrophic metabolism and biomass production rates (Hosen et al., 2014; Wu et al., 2018). Surface water
from the sink-rise system had the highest concentrations of DOC (Table 2) and POC (Fig. 2 a), but according to high HIX
and low FI and BIX values (Table 2), suggest the DOC origin was terrigenous and had low quality. Although groundwater
discharged from Head Spring and Devil´s Eye Spring had DOC and POC concentrations similar to those at Madison Blue
Spring, DOC quality in the latter reversing spring was intermediate to values for River Rise and River Sink (Table 2).
Bioavailability of the microbially derived, protein-like DOC in the Ichetucknee springs (Flint et al., 2023) was confirmed by
the relatively high rates of cell-specific production observed in these samples (Table 2). The $\delta^{13}C_{POC}$ data also provided
evidence for differences in the carbon sources available (Fig. 2 b), with the isotopically lightest $\delta^{13}C_{POC}$ from Head Spring



coinciding with the shortest community double time observed (14 h) among the groundwaters and surface waters sampled (Table 2).

### 4.3 Microbial respiration

The rates of oxygen consumption in oligotrophic groundwaters discharged from the Ichetucknee and reversing springs (0.1
to 0.2 mg $L^{-1}$ $d^{-1}$) were within the range observed in the DOC-rich waters of the sink-rise system during low flow (Table 2). Significantly lower values occurred only in suboxic groundwater discharging from Devil's Eye Spring and Madison Blue Spring 1 d after it transitioned from negative to positive discharge (Fig. 4a). The low rates of oxygen consumption and DIC production observed in samples from Madison Blue Spring immediately following a reversal event may represent a temporal biogeochemical response of reversing spring systems. In the organic-rich waters at the sink-rise system, the highest rates of
oxygen consumption were detected under high flow conditions, whereas the highest rates of DIC production were during low flow (Fig. 4; Table 2) and when there was a substantial contribution of groundwater to the conduits (Flint et al., 2023). Transitions in hydrological stage in the sink-rise system during river flooding led to increases in DO concentrations (from an average of 1.6 and 3.2 mg $L^{-1}$ at Rive Rise and from an average of 4.18 and 5.29 at the River Sink at low and high flow, respectively; Table 1) and DOC concentrations, but higher values of HIX suggest the DOC was of poorer quality (Table 2).
While incomplete oxidation of lower quality DOC at high flow is possible, reduced oxygen concentrations during low flow could enable additional microbial sources of DIC from fermentative and anaerobic respiratory metabolisms. In particular, waters discharged at River Rise contain supersaturated concentrations of dissolved $N_2O$ that has been generated via denitrification, and $N_2O$ concentrations are consistently higher under low flow conditions (Flint et al., 2023). Hence, a component of DIC produced at River Rise, as well as in the $N_2O$-saturated discharge of Madison Blue Spring (Flint et al.,
2021), has likely originated from heterotrophic denitrification in the groundwater.

RQ values based on DIC production and oxygen consumption rates (0.35 to 4.85; Table 2) deviate from the generally assumed range of 0.8 to 1.2 that is based on the stoichiometry of oxidation for specific organic compounds. However, they are commensurate to the wide range of values that have been reported for natural bacterial assemblages in freshwater
ecosystems (0.25 to 4.6; Berggren et al., 2012). At the sink-rise system, the increase of RQ from < 0.6 at high flow to 2.66 and 4.85, respectively, at low flow implies a carbon source transition to highly oxidized, low molecular weight organic acids (Berggren et al., 2012; Allesson et al., 2016; Hilman et al., 2022) and/or an effect of supplementary DIC sources from anaerobic metabolisms (Flint et al., 2021, 2023). Decreasing methane concentrations and enrichment of $\delta^{13}CH_4$ from River Sink to River Rise indicates there is a contribution from methanotrophy (Oberhelman et al., 2023), which has a theoretical
RQ of 0.5 (Bastviken et al., 2008) and may partially explain the low RQ values observed during high flow (Table 2). RQ values for Madison Blue Spring (1.23 and 0.62) were based on data collected 1 and 94 d, respectively, after a period of reversal, supporting the contention that increased residence time in the aquifer leads to shifts in microbial physiology and carbon consumption.



To compare the empirical oxygen consumption rates with estimates based on mixing of groundwater with atmospheric gases
(i.e., OUR), we assumed that water enters the aquifer saturated with atmospheric oxygen, constant rates of oxygen
consumption during the subsurface residence time, and aerobic respiration was the only oxygen sink. For springs discharging
the oldest groundwater (Head Spring and Devil's Eye Spring), OUR grossly underestimates the observed oxygen
consumption rate by 433- and 180-fold, respectively (Table 2). OUR values for River Rise are more congruent with

measured rates and overestimate oxygen consumption by 4- and 16-fold during low and high flow, respectively (Table 2). A
"dramatic" decrease in the DO concentrations of groundwater discharged by springs in this region has been documented
since the 1970s (Heffernan et al., 2010). Although the biogeochemical basis for the decrease in DO is not well understood,
the temporal trend coincides with increasing $NO_3^-$ concentrations that have also been observed in spring discharge over the
last ~70 years (Hornsby et al., 2004; Munch et al., 2006). Elevated levels of reactive nitrogen species have been implicated

in enhancing $N_2O$ production in the UFA (Flint et al., 2021), but it is currently not know if nitrogen eutrophication of the
groundwater may also be enhancing microbial metabolic rates in the subsurface.

**4.4 Heterotrophic production and growth**

High correlation of heterotrophic production rates with DOC concentration (r = 0.86, p < 0.05) and quality (HIX, r = 0.86;
BIX and FI, r = -0.85; p < 0.05) support that both the availability and characteristics of DOC influenced growth of the

surface water and groundwater communities we investigated. In the organic matter- and DO-rich surface waters of River
Sink, biomass was produced at rates exceeding those reported for the Amazon River (28 µg C $L^{-1}$ $d^{-1}$; Benner et al., 1995)
and were comparable to values for high organic matter river systems such as the Columbia River estuary (3,120 to 114,240
µg C $L^{-1}d^{-1}$; Herfort et al., 2017). Groundwater contributions to the sink-rise system are minimized when discharge rates are
high (> 15 $m^3$ $s^{-1}$), and while these hydrological conditions tend to increase DO and DOC concentration, the terrigenous-

derived DOC was generally of lower quality than that observed when groundwater is entering the conduits (Flint et al.,
2023). Nevertheless, rates of heterotrophic production, cell reproduction, and oxygen consumption in the sink-rise system
decreased during low flow conditions (Tables 1 and 2). One possibility is that metabolism becomes limited by electron
acceptor availability, similar to biogeochemical scenarios described for the conduits of Madison Blue Spring during periods
of reversal (Brown et al., 2014, 2019). Though BGEs as high as 80% have been reported in aquatic systems (Del Giorgio and

Cole, 1998; Eiler et al., 2003), the values inferred from the ³H-leucine and DIC data for the River Sink and River Rise (> 92
%) are so high they suggest our experimental approach underestimated microbial respiration or overestimated heterotrophic
production in these samples. The former possibility is less likely given that we expect additional $CO_2$ formed via anaerobic
metabolisms in these waters, which should result in an overestimate of aerobic respiration based on ΔDIC. Very similar and
high BGEs values (> 90 %) are also derived from the ³H-thymidine incorporation data, leading us to conclude that both of

these methods overestimated heterotrophic production (e.g., see Giering and Evans, 2022) in samples from the sink-rise
system.



Heterotrophic production was lower in groundwater discharging from the reversing and Ichetucknee springs than surface waters from the Santa Fe River, and four to eight orders of magnitude higher than that reported for a karstic aquifer in the

European Alps (Wilhartitz et al., 2009) and oligotrophic groundwater from sand and gravel aquifers (Hofmann and Griebler, 2018; Karwautz et al., 2022). Temperature-dependent reaction kinetics alone do not fully explain these differences, assuming standard responses for reaction kinetics (i.e., $Q_{10}$ values of 2 to 3; Gillooly et al., 2001) when comparing the warmer groundwater temperatures of ~22 °C in the UFA with temperatures of 4 and 12 °C, in the Alpine and sand and gravel aquifers, respectively; (Hofmann and Griebler, 2018; Karwautz et al., 2022). Multiple mechanisms may explain the observed

higher rates of microbial productivity in UFA groundwater, and several observations make a strong case for higher bioavailability of organic matter. Higher availability of metabolizable organic matter in the UFA groundwaters samples was substantiated by the much shorter inferred doubling times (from 0.58 d for Head Spring to 43 d for Madison Blue Spring; Table 2) when compared to those for alpine karst aquifer (712 d, groundwater residence time of ~22 years; Wilhartitz et al., 2009) and oligotrophic groundwater (533 d; Karwautz et al., 2022) systems. In addition, microbes discharged from Head

Spring had the highest rate of leucine relative to thymidine incorporation, potentially representing a response to growth limiting conditions (e.g., Church 2008). Nevertheless, these populations had rates of specific heterotrophic production (0.2607 pmol C cell$^{-1}$ h$^{-1}$) that were twice values for surface waters (Table 2), and 2 to 5 orders of magnitude higher than those for groundwater of comparable residence time (Wilhartitz et al., 2009). Lastly, the DOC examined in telogenetic alpine aquifers was generally of lower quality (i.e., average FI of 1.6; Harjung et al., 2023) relative to that we have documented in

the Ichetucknee springs (average FI of 1.78).

When Withlacoochee River level is low, groundwater discharging to Madison Blue spring run has a similar composition and DOC concentration to that in the nutrient-limited Ichetucknee springs (Table 1), yet a large disparity was observed in reproduction rates (Table 2). During the period of study, Madison Blue Spring reversed flow eight times; events that

transported electron donor- and acceptor-rich river water directly into the subsurface. Large volumes of water can recharge during reversals. During a 7.5 d reversal in 2009, an estimated ~$5.8 \times 10^4$ m$^3$ of river water recharged the conduits of Madison Blue Spring (Gulley et al., 2011). Increases in bioreactive solute concentration during reversals coincide with transient microbial blooms that have been observed by cave divers in the typically nutrient limited environment of the conduits (Gulley et al., 2011, 2013; Brown et al., 2014). Based on these observations, we initially hypothesized that

metabolic rates of subsurface communities in reversing springs would be intermediate to those of the Ichetucknee springs and sink-rise system. Though respiration rates (Fig. 4) and DO, DOC, and POC concentrations (Table 2; Fig. 2 a) at Madison Blue Spring were similar to other UFA groundwaters, the cell carbon production rates were at least 2-fold lower than those for the Ichetucknee springs (Fig. 5; Table 2) and do not provide support for our initial hypothesis. The heterotrophic production and respiration data from Madison Blue Spring produced BGEs (32 % and 39 %) that are lower than other sites

(Table 2) and comparable to values from oligotrophic portions of the ocean (Del Giorgio and Cole, 1998) and subsurface



aquatic systems in Antarctica (Vick-Majors et al., 2016). Low BGE at Madison Blue Spring could indicate uncoupling between catabolism and anabolism, which is consistent with low leucine to thymidine ratios (Table 2) that imply its groundwater community was investing most of their energy flow in maintenance metabolism rather than growth (e.g., Chin-Leo and Kirchman, 1990; Wos and Pollard, 2012). Previous investigations at Madison Blue Spring have shown altered

groundwater chemistry for a period of ~1 month after each reversal (Brown et al., 2014). Considering this, the intervals we sampled (e.g., Figure 5 data were collected 52 to 107 d after a transition from negative to positive discharge) might have been deemed sufficient to avoid the transient geochemical effects of flow reversal. However, our data make clear that further studies that include high temporal resolution sampling are needed to constrain the legacy effects of spring flow reversal on the availability of nutrients and redox couples for microbial metabolism in the groundwater.

**5 Conclusion**

Groundwater discharged from unconfined portions of the UFA contained low standing stocks of microbes that divided and produced cell carbon at rates greatly exceeding those previously documented for oligotrophic aquifers. Biogeochemical activity in the aquifer is enhanced due to extensive exchange of organic matter- and DO-rich surface water with groundwater across the karst landscape. Comparably high rates of cell specific metabolism and reproduction in the groundwater ─some

of which exceeded values observed for surface waters─ indicate the presence of readily oxidizable and assimilable organic carbon sources. Since labile pools of organic carbon in surface waters would not be expected to persist for the timeframes necessary to be transported with the oldest groundwaters, the results indicate a subsurface source of organic matter that may be supplied via a combination of chemoautotrophy, secondary production, and degradation of necromass (e.g., Geesink et al., 2022). Over recent decades, alarming groundwater quality trends have raised concerns about the health of Florida's

springs, including decreased rates of discharge (Florida Springs Institute, 2018), increased abundance of reactive nitrogen species (Katz et al., 2001; Katz, 2004; Katz et al., 2009; Denizman, 2018), and lower DO concentration in the groundwater and spring runs (Heffernan et al., 2010). Similar environmental changes are occurring globally, and thus, evaluations of mechanisms controlling microbial processes in the UFA may be useful to provide insights in other settings. An improved understanding of microbial biogeochemical activities affecting UFA groundwater quality is essential for developing

strategies to mitigate these challenging environmental issues and manage this vital natural resource under changing climate and land-use regimes. In addition to providing a basis for future studies that decipher the sources of organic matter driving biogeochemical processes in the UFA, the relationships we have documented among microbial biomass, physiology, and hydrogeochemistry provide an important case study about groundwater microbial communities that may be compared with conditions in different geological and environmental contexts of global karst landscapes.



**Data availability**

The DOC concentration data were previously published by Flint et al., (2021). DOC quality data are available at hydroshare: http://www.hydroshare.org/resource/a876020b85d6413f8486c57dc0b0e3bf. All other data collected and analyzed in this study are available at: http://www.hydroshare.org/resource/e8e4994bb1a740d8bb3fd65acf342cb6.

**Author contributions**

ABS: conceptualization, methodology, validation, formal analysis, investigation, data curation, writing – original draft, and visualization. MKF: methodology, formal analysis, investigation, data curation, and writing – review and editing. JCE: methodology, formal analysis, investigation, and writing – review and editing. JBM: conceptualization, resources, writing – review and editing, supervision, and funding acquisition. BCC: conceptualization, methodology, formal analysis, resources, writing – original draft, supervision, project administration, and funding acquisition.

**Competing interests**

The authors declare that they have no conflict of interest.

**Acknowledgements**

This study was supported by funding from the University of Florida's Biodiversity Institute (Seed grant #020518 to BCC and JBM; graduate fellowship to ABS) and a Fulbright scholarship (to ABS). Partial support was also provided by the Institute of Food and Agricultural Sciences at the University of Florida. Research at springs in North Central Florida was conducted under the Florida Department of Environmental Protection permits 06281812, 03211912, 07092012, 07162112A, and 08122212, and we are indebted to Christine Housel for her assistance with permit acquisition. We thank Patricia Spellman, Robert Sharping, and Jason Gulley for discussions; Quincy Faber for assisting with collecting the ATP data; and Kelenna O. Irving, Madison Tharp, Victoria Cassady, Arianna Insenga, Rachel Pinsky, and Katelyn Palmer for their assistance with field work.

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
