# Peer review of "Effects of surface water interactions with karst groundwater on microbial biomass, metabolism, and production"

_EGUsphere, 2024_

## Referee Comment (RC1)

**Effects of surface water interactions with karst groundwater on microbial biomass, metabolism, and production**

Adrian Barry-Sosa, Madison K. Flint, Justin C. Ellena, Jonathan B. Martin, Brent C. Christner

**Review**

**Summary and general comments**

The manuscript from Barry-Sosa et al. represents a comprehensive analysis of surbsurface biogeochemistry within the karstic Upper Floridan Aquifer that includes sites across a range of surface water and groundwater mixing. The dataset is extremely comprehensive (& cool!), including a wide variety of analysis techniques that expansively assess water chemistry (temperature, pH, conductivity, DO, OM quality using EEMs, etc.), microbiome characteristics (cell counts, biomass), and microbiome activity (respiration calculations via incubations, heterotrophic production with $^3$H-leucine and -thymidine). The vast variety of tools used makes this dataset incredibly unique. Overall, I believe that the data are solid and that this is an impactful contribution to the research community for assessing how surface water-groundwater interactions in karst systems influence biogeochemistry. However, with all this data, I struggle following the broader story and tracking whether the data aligns with the hypothesis stated in the introduction and think there are some major improvements that must be made to refine the manuscript for publication.

**Major comments**

You mention regional groundwater in L266 but I don't think its mentioned again. Because all the springs & river samples provide this gradient of SW-GW mixing (nicely introduced in L9-L10 of abstract) across the region, it would be nice to include data more prominently from a true regional groundwater sample as a sort of end member comparison. In addition, having a surface water sample as the opposite end member (maybe the Santa Fe Sink & Rise system during high river discharge) could be cool. If you had data from these, you could use your conductivity data as a "conservative tracer" to quantify actual SW-GW mixing at each of your springs which would provide a nice backdrop for all of the presented data. See "Figures" comments and others regarding the necessity for displaying the gradient in SW-GW mixing more prominently throughout, and this could be a cool way to do so.

**Introduction:** Though the introduction is very well-written (I felt like I could easily understand the importance & relevance as someone that studies SW-GW mixing but not in karst systems), I feel that the last paragraph needs to be reworked – maybe into 2 separate paragraphs – so that the reader can fully grasp previous work and the importance of this work in the UFA system. I say a new paragraph because there seems to be a lot of previous research on the UFA that is briefly mentioned in L61-65. Because of the amount of work done here, I think it would help to have a paragraph prior to this discussion the previous research done here and what is known about these springs. I'd especially touch on the differing discharges and what is known about how this alters organic matter (mentioned in L61) because this is directly relevant to this manuscript. I'd then end the introduction with a final paragraph that succinctly states what you are doing

that is new, your hypothesis, how you addressed it, and briefly what you found. I make these recommendations because I feel that this manuscript relies heavily on the readers understanding of the sites (which might be able to be addressed, see other major comments) and I leave the introduction not feeling like I have the full background to follow the significance of the dataset through the results & discussion.

**Figures**: There is a **lot** of data presented in this manuscript and I had trouble understanding or following what the broader story was, especially in the results section which is very lengthy. I tend to go to manuscript figures when I feel I can't follow the overarching story, but the figures are not super helpful if you don't fully understand the relevance of the different sites. Because of this, I had to keep jumping back to Section 2.1 to remind myself of the differences between the sites. I understand that the relevance here is the backdrop of different groundwater-surface water mixing between the different springs & river sink-rise system, but those differences are only primarily presented in the Section 2.1 and Section 3.1 (& yes, sprinkled in here & there). I feel that underlying differences in GW-SW mixing between sites need to be much more prominently displayed in the figures and that the presenting of data needs to be changed for readers to be able to follow the main story without jumping back to Section 2.1. The entire relevance of the different sites is the altered GW-SW regimes but this is completely lost in how the data is currently presented. Overall, editing the figures so that the readers can more easily understand the takeaway from the dataset would help in following the overarching findings that shifts in GW-SW mixing in these karst systems impacts biogeochemistry and greatly strengthen the manuscript!

A few potential suggestions below:
-Adding more information to all boxplot figures, perhaps grouping and labeling sites by discharge (e.g., Peacock Springs, Madison Blue Spring, Little River Spring, & Gilchrist Blue Springs all labeled as reversing springs and placed together on boxplots).
-I think you have discharge information for each site (at least the majority?, Table S1), for some of the data it would be interesting to plot against discharge with points shaped by site.

Another note on figures: In addition to thinking of other ways to display your data to include the different GW-SW regimes, I feel that more information needs to be added to the figures as presented. For example, the axes of all figures (besides the site figure, Fig. 1) need to include more than just the units (ex: Fig. 2a needs to say POC concentration (mg C L-1)). I also think the manuscript needs to be more consistent with the use of acronyms or not. Each figure uses the site acronyms (which I get as the site names can be lengthy), but the text largely uses the full site names. Make sure the manuscript is consistent throughout: either exclusively use acronyms or exclusively use the full site names, across the figures and text.

**Results:** The results section is very long – which I get as there is a lot of data here – but I recommend condensing. Of course, still comprehensively present the data but the manuscript doesn't need to explicitly state every data point! I'd also try to rework so that again the data is presented with the backdrop of hydrology regimes. The first paragraph

of Section 3.1 is awesome and super helpful but with this many sites it's hard to remember all the different hydrology regimes, especially as someone that has never worked in the UFA. For example, in L278-280, the manuscript presents which sites have the heaviest δ13C value for POC and the variation in the dataset. I'd refine this so that it is framed with the hydrology (e.g., "the heaviest δ13C value for POC was observed at MB that had a [higher/lower?] influence of GW than these other sites with [less/more] GW influence).

**Discussion:** I love Section 4.1! Super helpful in framing the discussion and reaffirming the significance to the reader. I do feel that the discussion is also very lengthy, and should maybe be reformatted so, instead of it sectioned by the different datasets, format it by the different key findings. Some of the points in the discussion seemed to be more fitting for a results section (e.g., L523-524, the correlation analyses and p-values fit more in a results discussion). I recommend condensing the discussion and focusing mainly on what the data means: does it agree with previously published work on UFA, what are the *key* findings, continuously connecting findings to hypothesis, include significance of results. Again, there is just a lot here and I struggled coming away understanding the key findings of this work!

**Conclusion**: Super well-written, but I feel that some of this could instead be in the discussion. In my experience, the discussion should include significance of findings and the conclusion includes an overview of what you did & found with suggestions for future work.

**Minor comments**
L40: Remove extra Jin in citation. (Jin et al., 2014)

L70: Define what a river sink-rise system is. I work in alpine streams and have never heard this term!

L72-L75: Mentioned in major comments, be consistent with acronyms throughout. I think this is the only time acronyms are used in the text, but are always used in the figures.

L145: Add a clarifier to what entails "humic characteristics", especially as folks have moved away from the terms "humics" and "humification" in the OM world. Something like: "The Humification Index (HIX) indicates degree of polycondensation where higher HIX values are more indicative of lower H/C ratios and higher molecular weights". "Humic characteristics" isn't super informational or specific!

L179: Define ATP as its the first time its used!

L253-L55: No suggestion just that this is cool!! And a good presentation of the data with the included background of the hydrology here!

L266: You mention regional groundwater here, it would be helpful to include throughout the reference of this regional groundwater sample to provide some sort of end member to compare the data! See major comments for more details here.

Table 2: Missing the first row that you have in Table 1 that includes the different groups.

L470: Try to use better descriptors than "quality" here. The use of quality isn't helpful to the reader, you can use quality but include more like: "….had low quality (e.g., higher H/C ratios).

L523: This correlative relationship would be nice to include in a figure.

---

## Author Comment (AC1)

**Response to Reviewer 1**
Barry-Sosa et al., "Effects of surface water interactions with karst groundwater on microbial biomass, metabolism, and production"

**R1.C1:** "*You mention regional groundwater in L266 but I don't think its mentioned again. Because all the springs & river samples provide this gradient of SW-GW mixing (nicely introduced in L9-L10 of abstract) across the region, it would be nice to include data more prominently from a true regional groundwater sample as a sort of end member comparison. In addition, having a surface water sample as the opposite end member (maybe the Santa Fe Sink & Rise system during high river discharge) could be cool. If you had data from these, you could use your conductivity data as a "conservative tracer" to quantify actual SW-GW mixing at each of your springs which would provide a nice backdrop for all of the presented data. See "Figures" comments and others regarding the necessity for displaying the gradient in SW-GW mixing more prominently throughout, and this could be a cool way to do so.*"

> **Author's response:** Based on this comment, we realize that "regional groundwater" is not the best word choice for this description. Hence, we propose replacing it with "matrix water" (i.e. water stored and circulating through the aquifer rock porosity).
>
> Using a conservative tracer to quantify the degree of SW-GW mixing is an excellent idea. However, the main challenge in applying this approach to our data is that we lack a true groundwater endmember for the River Sink-Rise system, which has at least two distinct groundwater sources that are difficult to discern from each other. This together with surface water mixing causes wide variability in the matrix water geochemistry as indicated by measurements at monitoring wells in the area (See Table 1, Moore et al., 2009, J. Hydrol., 376, 443–455, https://doi.org/10.1016/j.jhydrol.2009.07.052). In addition, separating these three endmembers would require more solute data than we have for this project.

**R1.C2:** "*Though the introduction is very well-written (I felt like I could easily understand the importance & relevance as someone that studies SW-GW mixing but not in karst systems), I feel that the last paragraph needs to be reworked – maybe into 2 separate paragraphs – so that the reader can fully grasp previous work and the importance of this work in the UFA system. I say a new paragraph because there seems to be a lot of previous research on the UFA that is briefly mentioned in L61-65. Because of the amount of work done here, I think it would help to have a paragraph prior to this discussion the previous research done here and what is known about these springs. I'd especially touch on the differing discharges and what is known about how this alters organic matter (mentioned in L61) because this is directly relevant to this manuscript. I'd then end the introduction with a final paragraph that succinctly states what you are doing that is new, your hypothesis, how you addressed it, and briefly what you found. I make these recommendations because I feel that this manuscript relies heavily on the readers understanding of the sites (which might be able to be addressed, see other major comments) and I leave the introduction not feeling like I have the full background to follow the significance of the dataset through the results & discussion.*"

> **Author's response:** We agree with this suggestion and propose to divide the final introduction paragraph as recommended. While the site description section provides explicit detail on existing information for each site, we intend to briefly highlight the most relevant information in the introductory paragraph of the revised manuscript. We propose to describe how prior research conducted at many of our sites has shown residence time in the aquifer and surface-groundwater interactions are connected to the levels of dissolved oxygen, organic carbon, and microbes in the groundwater, but that the effects of groundwater hydrogeochemistry on microbial physiology and

production has not been studied. In the revision, we will also edit the final paragraph of the introduction to succinctly state our experimental hypotheses and the significance of our results.

**R1.C3:** *"I feel that underlying differences in GW-SW mixing between sites need to be much more prominently displayed in the figures and that the presenting of data needs to be changed for readers to be able to follow the main story without jumping back to Section 2.1. The entire relevance of the different sites is the altered GW-SW regimes but this is completely lost in how the data is currently presented. Overall, editing the figures so that the readers can more easily understand the takeaway from the dataset would help in following the overarching findings that shifts in GW-SW mixing in these karst systems impacts biogeochemistry and greatly strengthen the manuscript!."*

**Author's response:** Thank you for this suggestion. We propose to rearrange the site order in Table 2 and Figs. 2 to 5 and supplementary figures S1, S2, S5 and S7 so that the three categories of springs are clearly indicated and the sites are maintained in the same order for consistency of presentation. As an example, we show the new presentation for a revision of Fig. 5:

[Figure]

**R1.C4:** *"I I feel that more information needs to be added to the figures as presented. For example, the axes of all figures (besides the site figure, Fig. 1) need to include more than just the units (ex: Fig. 2a needs to say POC concentration (mg C L-1))."*

**Author's response:** We agree with this suggestion and propose to add the following descriptive information to the y-axis labels in the revision:
Figure 2 (a): "POC (mg C L$^{-1}$ d$^{-1}$)"
Figure 3 (b): "Cell-based biomass (ng C L$^{-1}$)"
Figure3 (c): "ATP-based biomass (ng C L$^{-1}$)"
Figure 3 (d): "ATP/cell (zmol)"
Figure 4 (a): "O$_2$ consumption rate (mg O$_2$ L$^{-1}$ d$^{-1}$)"
Figure 4 (b): "DIC production rate (µg C g$^{-1}$ d$^{-1}$)"
Figure 5: "Heterotrophic production (µg C g$^{-1}$ d$^{-1}$)"

**R1.C5:** *"I also think the manuscript needs to be more consistent with the use of acronyms or not. Each figure uses the site acronyms (which I get as the site names can be lengthy), but the text largely uses the full site names. Make sure the manuscript is consistent throughout: either exclusively use acronyms or exclusively use the full site names, across the figures and text."*

> **Author's response:** When drafting the manuscript and figures, we had an active debate on whether or not to use acronyms through the manuscript. Ultimately, we decided that the acronym heavy text would be extremely unfriendly to the reader. We drafted figures that replaced the acronyms with full names, but the limited space available in the multi-sample plots did not allow the full names to be adequately displayed. To address this comment, we are proposing to use the full spring names in the text, keep the acronyms in the figure axes, but define the acronyms for each figure in the legend and/or figure caption. We believe this will make the text easily readable without an acronym key and that the addition information provided in the legend will make deciphering the sample designations easier. Please also see our response to R1.C3, which will also assist with differentiating among the site types.

**R1.C6:** *"The results section is very long – which I get as there is a lot of data here – but I recommend condensing. Of course, still comprehensively present the data but the manuscript doesn't need to explicitly state every data point! I'd also try to rework so that again the data is presented with the backdrop of hydrology regimes. The first paragraph of Section 3.1 is awesome and super helpful but with this many sites it's hard to remember all the different hydrology regimes, especially as someone that has never worked in the UFA. For example, in L278-280, the manuscript presents which sites have the heaviest δ13C value for POC and the variation in the dataset. I'd refine this so that it is framed with the hydrology (e.g., "the heaviest δ13C value for POC was observed at MB that had a [higher/lower?] influence of GW than these other sites with [less/more] GW influence)."*

> **Author's response:** This criticism largely concerns style and we believe that a large reduction in the results section would eliminate valuable information that is of interest to many readers. However, this feedback has prompted us to identify areas in the results that can be streamlined to improve flow and the presentation. This includes the section on POC mentioned (lines 278-280) as well as content in lines 335-345, which describes the ATP biomass measurements in explicit detail. We also agree that revising sections of the results to emphasize the influence of surface water and groundwater at the sites discussed would assist the reader in evaluating the effects of hydrology on the various biogeochemical variables discussed.

**R1.C7:** *"I do feel that the discussion is also very lengthy, and should maybe be reformatted so, instead of it sectioned by the different datasets, format it by the different key findings. Some of the points in the discussion seemed to be more fitting for a results section (e.g., L523-524, the correlation analyses and p-values fit more in a results discussion). I recommend condensing the discussion and focusing mainly on what the data means: does it agree with previously published work on UFA, what are the key findings, continuously connecting findings to hypothesis, include significance of results."*

> **Author's response:** Thank you for this feedback to improve the discussion. We agree that the section headers in the initial submission could be revised to be more informative and appropriate for the discussion. Therefore, we propose modifying the following section headers as follows:

| Section heading – initial submission | Section heading - revision |
|---|---|
| Groundwater and surface water mixing in karst landscapes | Importance of groundwater and surface water mixing in karst landscapes |
| Abundance of microbes, biomass, and organic carbon | Effects of groundwater residence time on microbes, biomass, and organic carbon |

| Microbial respiration | Hydrology and DOC influences on microbial respiration |
| Heterotrophic production and growth | Heterotrophic production and growth do not scale with groundwater age |

Similar to comment R1.6, the criticism on the length of the discussion is about style, but we do agree that the discussion can be improved by better highlighting the key findings and their significance, as well as eliminating text that is more suitable for the results such as lines 523-524.

**R1.C8:** "Conclusion: *Super well-written, but I feel that some of this could instead be in the discussion. In my experience, the discussion should include significance of findings and the conclusion includes an overview of what you did & found with suggestions for future work.*"

**Author's response:** To address this comment, we will move and integrate the text from lines 593-597 into the first paragraph of the discussion (lines 438-448).

**R1.C9:** "*L40: Remove extra Jin in citation. (Jin et al., 2014)*"

**Author's response:** Thank you for pointing out this typographical error, which we have noted to correct in the revised manuscript.

**R1.C10:** *"L70: Define what a river sink-rise system is."*

**Author's response:** We agree with this suggestion and propose to parenthetically add the following definition to this sentence: "(i.e., a swallet or opening where a river recharges water-filled caves and discharges to the surface at a point downstream)".

**R1.C11:** *"L72-L75: Mentioned in major comments, be consistent with acronyms throughout. I think this is the only time acronyms are used in the text, but are always used in the figures."*

**Author's response:** Please see our response to R1.C5.

**R1.C12:** *"L145: Add a clarifier to what entails "humic characteristics", especially as folks have moved away from the terms "humics" and "humification" in the OM world. Something like: "The Humification Index (HIX) indicates degree of polycondensation where higher HIX values are more indicative of lower H/C ratios and higher molecular weights". "Humic characteristics" isn't super informational or specific!"*

**Author's response:** We agree and will revise this section using the phrasing suggested.

**R1.C13:** *"L179: Define ATP as its the first time its used!"*

**Author's response:** Agreed.

**R1.C14:** *"L266: You mention regional groundwater here, it would be helpful to include throughout the reference of this regional groundwater sample to provide some sort of end member to compare the data! See major comments for more details here."*

**Author's response:** Please see our response to R1.C1.

**R1.C15:** "*Table 2: Missing the first row that you have in Table 1 that includes the different groups."*

**Author's response:** The revised Table 2 will organize and separate the springs identically to that shown for Table 1.

**R1.C16: "***L470: Try to use better descriptors than "quality" here. The use of quality isn't helpful to the reader, you can use quality but include more like: "….had low quality (e.g., higher H/C ratios)."*

**Author's response:** Thank you for this suggestion. We will revise the text to indicate that low quality carbon is inferred by high HIX and low FI and BIX values.

**R1.C17: "***L523: This correlative relationship would be nice to include in a figure***"**

**Author's response:** Agreed. A plot showing the correlation of heterotrophic production with DOC concentration and quality (HIX, FI, and BIX values) will be added as a supplementary figure to the revised manuscript.

---

## Author Comment (AC2)

**Response to review Reviewer 2**
Barry-Sosa et al., "Effects of surface water interactions with karst groundwater on microbial biomass, metabolism, and production"

**R2.C1:** "*In Table 1, some instances of "N.d." were written as "N.d" in the row for "DIC".*

   **Author's response:** Thank you for pointing out this typographical error. We will correct these discrepancies in Table 1 of the revised manuscript.

**R2.C2:** "*the note below Table 1 for "Sp. Cond" does not match the values in the table.*"

   **Author's response:** We are a little confused by this comment. If this refers to "Sp. Cond." in the table appearing as "Sp. Cond" in the legend, we will add a period to the latter in the revision.

**R2.C3:** "*Table 2 contains two variations of "Not applicable", mentioned as "N.a" or "N.a."*".

   **Author's response:** Thank you for pointing out this discrepancy, which we can easily correct in the revision.

**R2.C4:** "*The figures appear blurry, and the color of the graph's axes could be made darker to enhance clarity for readers.*"

   **Author's response:** We agree with this suggestion and will also use higher resolution images for the figures in the revised manuscript.

**R2.C5:** "*In line 351-352, please provide accurate data or a figure to support the statement regarding "ATP-based biomass concentrations in samples from River Rise at low flow being significantly higher than values at high flow and approximately 3-fold lower than those observed for River Sink at low flow."*".

   **Author's response:** In the initial submission, we neglected to point out that this statement is supported by the data in Fig. 3 c, which we intend to include in the revised text.

**R2.C6:** *Line 357-359 refers to "Fig. 2c," but there is no such figure mentioned in the article.*

   **Author's response:** Thank you for pointing out this oversight, which will be corrected in the revision.

**R2.C7:** "*In line 390-392 and line 407-408, the authors state that the incorporation rate of 3H-thymidine was much higher from Madison Blue Spring. Therefore, it may be more appropriate to use the 3H-thymidine incorporation rate when evaluating doubling times at Madison Blue Spring.*".

   **Author's response:** More rapid generation times were inferred from the $^3$H-thymidine at all sites, but for Madison Blue Spring, they were much more rapid (7 to 30 h versus 232 to 1028 h for the $^3$H-leucine incorporation data; Barry Sosa, PhD thesis, University of Florida, 2023). Additionally, we observed very low ratios of leucine to thymidine incorporation at Madison Blue Spring, which were ~10-fold lower than those observed at the other sites (Table 2). This coupled with low BGE values implied uncoupling between catabolism and anabolism and/or that the community was investing most of its energy flow in maintenance metabolism rather than growth (see discussion in lines 575-578). For these reasons and those described in lines

403-405, we believe the ³H-leucine data are the most appropriate for estimating doubling times and bacterial production to make comparisons among sites. Please note that ³H-thymidine incorporation rates are provided in the supplementary materials (Figure S7), which allow anyone seeking to makes direct comparisons to calculate bacterial production and doubling rates from these data.

**R2.C8:** *"In line 460, please attempt to explain why ATP contents are high at Devil's Eye Spring."*

**Author's response:** The very high per cell ATP contents inferred in groundwater from Devil's Eye Spring are surprising and remain unexplained. The simplest possible explanations are that 1) the groundwater community at Devil's Eye has higher amounts of biomass/ATP, 2) is producing ATP relatively more rapidly than the communities at other sites, or 3) the excesses are due to low anabolic rates that consume ATP slowly and allow produced ATP to accumulate in the cells. Given that the first two options are not supported by our data (see lines 462-464), the latter explanation (#3) is the most likely. We propose to add a few sentences in the revision that explore these possibilities and state this working hypothesis.

---

## Author Response (AR1)

**Changes to originally submitted manuscript**
Barry-Sosa et al., "Effects of surface water interactions with karst groundwater on microbial biomass, metabolism, and production"

Please note that when describing changes to the manuscript, line numbers refer to the revised manuscript. In the case of deleted lines and referee's comments, they refer to the originally submitted manuscript.

**R1.C1:** "*You mention regional groundwater in L266 but I don't think its mentioned again. Because all the springs & river samples provide this gradient of SW-GW mixing (nicely introduced in L9-L10 of abstract) across the region, it would be nice to include data more prominently from a true regional groundwater sample as a sort of end member comparison. In addition, having a surface water sample as the opposite end member (maybe the Santa Fe Sink & Rise system during high river discharge) could be cool. If you had data from these, you could use your conductivity data as a "conservative tracer" to quantify actual SW-GW mixing at each of your springs which would provide a nice backdrop for all of the presented data. See "Figures" comments and others regarding the necessity for displaying the gradient in SW-GW mixing more prominently throughout, and this could be a cool way to do so.*"

> **Author's response:** Based on this comment, we realize that "regional groundwater" is not the best word choice for this description. Hence, we propose replacing it with "matrix water" (i.e. water contained in matrix porosity).
>
> Using a conservative tracer to quantify the degree of SW-GW mixing is an excellent idea. However, the main challenge in applying this approach to our data is that we lack a true groundwater endmember for the River Sink-Rise system, which has at least two distinct groundwater sources that are difficult to discern without more solute data than we have for this project. This together with surface water mixing causes wide variability in the matrix water geochemistry as indicated by measurements at monitoring wells in the area (See Table 1, Moore et al., 2009, J. Hydrol., 376, 443–455, https://doi.org/10.1016/j.jhydrol.2009.07.052).
>
> **Changes to the manuscript:** The term "regional groundwater" has been replaced by "matrix water" (L278) and revised to include a definition (L278-279).

**R1.C2:** "*Though the introduction is very well-written (I felt like I could easily understand the importance & relevance as someone that studies SW-GW mixing but not in karst systems), I feel that the last paragraph needs to be reworked – maybe into 2 separate paragraphs – so that the reader can fully grasp previous work and the importance of this work in the UFA system. I say a new paragraph because there seems to be a lot of previous research on the UFA that is briefly mentioned in L61-65. Because of the amount of work done here, I think it would help to have a paragraph prior to this discussion the previous research done here and what is known about these springs. I'd especially touch on the differing discharges and what is known about how this alters organic matter (mentioned in L61) because this is directly relevant to this manuscript. I'd then end the introduction with a final paragraph that succinctly states what you are doing that is new, your hypothesis, how you addressed it, and briefly what you found. I make these recommendations because I feel that this manuscript relies heavily on the readers understanding of the sites (which might be able to be addressed, see other major comments) and I leave the introduction not feeling like I have the full background to follow the significance of the dataset through the results & discussion.*"

> **Author's response:** We agree with this suggestion and propose to divide the final introduction paragraph as recommended. While the site description section provides explicit details on existing

information for each site, we intend to briefly highlight the most relevant information in the introductory paragraph of the revised manuscript. In the revision, we will also edit the final paragraph of the introduction to succinctly state our experimental hypotheses and the significance of our results.

**Changes to the manuscript:** We revised the first paragraph to describe how prior research conducted at many of our sites has shown residence time in the aquifer and surface-groundwater interactions are connected to the levels of dissolved oxygen, organic carbon, and microbes in the groundwater, but that the effects of groundwater hydrogeochemistry on microbial physiology and production has not been studied (L61-67). We also revised the final paragraph of the introduction to specifically state the hypothesis of our study and the relevance of its findings (L69-77).

**R1.C3:** *"I feel that underlying differences in GW-SW mixing between sites need to be much more prominently displayed in the figures and that the presenting of data needs to be changed for readers to be able to follow the main story without jumping back to Section 2.1. The entire relevance of the different sites is the altered GW-SW regimes but this is completely lost in how the data is currently presented. Overall, editing the figures so that the readers can more easily understand the takeaway from the dataset would help in following the overarching findings that shifts in GW-SW mixing in these karst systems impacts biogeochemistry and greatly strengthen the manuscript!."*

**Author's response:** Thank you for this suggestion. We propose to rearrange the site order in Table 2 and Figs. 2 to 5 and supplementary figures S1, S2 and S8 so that the three categories of springs are clearly indicated and the sites are maintained in the same order for consistency of presentation.

**Changes to the manuscript:** The order of springs in Table 2 has been rearranged. In Figs 2 to 5 and supplementary figures S1, S2 and S8, the three spring categories are indicated and the spring order has been rearranged to be consistent.

**R1.C4:** *"I I feel that more information needs to be added to the figures as presented. For example, the axes of all figures (besides the site figure, Fig. 1) need to include more than just the units (ex: Fig. 2a needs to say POC concentration (mg C L-1))."*

**Author's response:** We agree with this suggestion.

**Changes to the manuscript:** We added the following labels to the figures mentioned below:
Figure 2 (a): "POC (mg C $L^{-1}$ $d^{-1}$)"
Figure 3 (b): "Cell-based biomass (ng C $L^{-1}$)"
Figure3 (c): "ATP-based biomass (ng C $L^{-1}$)"
Figure 3 (d): "ATP/cell (zmol)"
Figure 4 (a): "$O_2$ consumption rate (mg $O_2$ $L^{-1}$ $d^{-1}$)"
Figure 4 (b): "DIC production rate (µg C $g^{-1}$ $d^{-1}$)"
Figure 5: "Heterotrophic production (µg C $g^{-1}$ $d^{-1}$)"
Figure S8: "Heterotrophic production (µg C $g^{-1}$ $d^{-1}$)"

**R1.C5:** *"I also think the manuscript needs to be more consistent with the use of acronyms or not. Each figure uses the site acronyms (which I get as the site names can be lengthy), but the text largely uses the full site names. Make sure the manuscript is consistent throughout: either exclusively use acronyms or exclusively use the full site names, across the figures and text."*

**Author's response:** When drafting the manuscript and figures, we had an active debate on whether or not to use acronyms through the manuscript. Ultimately, we decided that the acronym heavy text would be extremely unfriendly to the reader. We drafted figures that replaced the acronyms with full names, but the limited space available in the multi-sample plots did not allow the full names to be adequately displayed. To address this comment, we are proposing to use the full spring names in the text, keep the acronyms in the figure axes, but define the acronyms for each figure in the legend and/or figure caption. We believe this will make the text easily readable without an acronym key and that the additional information provided in the legend will make deciphering the sample designations easier. Please also see our response to R1.C3, which will also assist with differentiating among the site types.

**Changes to the manuscript:** Removed the acronyms from L83 to 86 and added an acronym key in each figure caption.

**R1.C6:** "*The results section is very long – which I get as there is a lot of data here – but I recommend condensing. Of course, still comprehensively present the data but the manuscript doesn't need to explicitly state every data point! I'd also try to rework so that again the data is presented with the backdrop of hydrology regimes. The first paragraph of Section 3.1 is awesome and super helpful but with this many sites it's hard to remember all the different hydrology regimes, especially as someone that has never worked in the UFA. For example, in L278-280, the manuscript presents which sites have the heaviest δ13C value for POC and the variation in the dataset. I'd refine this so that it is framed with the hydrology (e.g., "the heaviest δ13C value for POC was observed at MB that had a [higher/lower?] influence of GW than these other sites with [less/more] GW influence)."*

**Author's response:** This criticism largely concerns style and we believe that a large reduction in the results section would eliminate valuable information that is of interest to many readers. However, this feedback has prompted us to identify areas in the results that can be streamlined to improve flow and the presentation. This includes the section on POC mentioned (lines 278-280) as well as content in lines 335-345, which describes the ATP biomass measurements in explicit detail. We also agree that revising sections of the results to emphasize the influence of surface water and groundwater at the sites discussed would assist the reader in evaluating the effects of hydrology on the various biogeochemical variables discussed.

**Changes to the manuscript:** We have deleted L252-253, L336, L337-338, L341-344, L388-390 and added phrasing in L288, L293, L322, L347, L360, L370, L388-389, L419-420, L429, L431, and L444 linking predominance of GW/SW with given hydrological conditions/locations.

**R1.C7:** "*I do feel that the discussion is also very lengthy, and should maybe be reformatted so, instead of it sectioned by the different datasets, format it by the different key findings. Some of the points in the discussion seemed to be more fitting for a results section (e.g., L523-524, the correlation analyses and p-values fit more in a results discussion). I recommend condensing the discussion and focusing mainly on what the data means: does it agree with previously published work on UFA, what are the key findings, continuously connecting findings to hypothesis, include significance of results.*"

**Author's response:** Thank you for this feedback to improve the discussion. We agree that the section headers in the initial submission could be revised to be more informative and appropriate for the discussion. Therefore, we propose modifying the following section headers as follows:

| Section heading – initial submission | Section heading - revision |
| --- | --- |
| Abundance of microbes, biomass, and organic carbon | Effects of groundwater residence time on microbes, biomass, and organic carbon |

| Microbial respiration | Hydrology and DOC influence microbial respiration |
| Heterotrophic production and growth | Effect of surface water-groundwater interactions on heterotrophic growth |

Similar to comment R1.C6, the criticism on the length of the discussion is about style, but we do agree that the discussion can be improved by better highlighting the key findings and their significance, as well as eliminating text that is more suitable for the results such as lines 523-524.

**Changes to the manuscript:** Discussion subheadings have been changed as described above. Text formerly in lines 523-524 has been moved to L433-436 in the results.

**R1.C8: "**Conclusion: *Super well-written, but I feel that some of this could instead be in the discussion. In my experience, the discussion should include significance of findings and the conclusion includes an overview of what you did & found with suggestions for future work."*

**Author's response:** To address this comment, we have moved and integrated text from lines 593-597 into the first paragraph of the discussion.

**Changes to the manuscript:** The text formerly in L593-597 has been moved to L465-468 in the discussion. Wording in L613-614 in the conclusion has been adjusted to account for the text removal.

**R1.C9: "***L40: Remove extra Jin in citation. (Jin et al., 2014)"*

**Author's response:** Thank you for pointing out this typographical error, which we have noted to correct in the revised manuscript.

**Changes to the manuscript:** The error has been corrected and the extra Jin et al 2014 citation has been removed.

**R1.C10:** *"L70: Define what a river sink-rise system is."*

**Author's response:** We agree with this suggestion and propose to parenthetically add the following definition to this sentence: "(i.e., a swallet or opening where a river recharges water-filled caves and discharges to the surface at a point downstream)".

**Changes to the manuscript:** A new sentence defining a river sink-rise system has been added in L86-87.

**R1.C11:** *"L72-L75: Mentioned in major comments, be consistent with acronyms throughout. I think this is the only time acronyms are used in the text, but are always used in the figures."*

**Author's response:** Please see our response to R1.C5.

**Changes to the manuscript:** See changes made in R1.C5.

**R1.C12:** *"L145: Add a clarifier to what entails "humic characteristics", especially as folks have moved away from the terms "humics" and "humification" in the OM world. Something like: "The Humification Index (HIX) indicates degree of polycondensation where higher HIX values are more*

*indicative of lower H/C ratios and higher molecular weights". "Humic characteristics" isn't super informational or specific!"*

**Author's response:** We agree and will revise this section using the phrasing suggested.

**Changes to the manuscript:** The term "humic characteristics" has been removed and the term HIX has been defined in L156-158.

**R1.C13:** *"L179: Define ATP as its the first time its used!"*

**Author's response:** Agreed.

**Changes to the manuscript:** The acronym has been defined on first use in L191.

**R1.C14:** *"L266: You mention regional groundwater here, it would be helpful to include throughout the reference of this regional groundwater sample to provide some sort of end member to compare the data! See major comments for more details here."*

**Author's response:** Please see our response to R1.C1.

**Changes to the manuscript:** See changes done in response R1.C1.

**R1.C15:** *"Table 2: Missing the first row that you have in Table 1 that includes the different groups."*

**Author's response:** The revised Table 2 organizes and separates the springs identically to that shown for Table 1.

**Changes to the manuscript:** The location order has been rearranged to match the order they appear on Table 1. A new organization mirroring that in Table 1 is used to categorized each site location by type.

**R1.C16: "***L470: Try to use better descriptors than "quality" here. The use of quality isn't helpful to the reader, you can use quality but include more like: "....had low quality (e.g., higher H/C ratios)."*

**Author's response:** Thank you for this suggestion. We have revised the text to indicate that low quality carbon is inferred by high HIX and low FI and BIX values.

**Changes to the manuscript:** We added a short statement to clarify the meaning of "low quality carbon in L491 describing that low quality carbon is inferred by high HIX and low FI and BIX values. In L294-295, the parameters used to define "high quality" organic matter are also described. In addition, we briefly state what we mean by carbon quality in the introduction in L69-70.

**R1.C17: "***L523: This correlative relationship would be nice to include in a figure***"**

**Author's response:** Agreed.

**Changes to the manuscript:** An additional figure (Fig. S7) has been added to the supplementary material showing the correlation of heterotrophic production with DOC concentration and quality (HIX, FI, and BIX values).

**Response to review Reviewer 2**

**R2.C1: "***In Table 1, some instances of "N.d." were written as "N.d" in the row for "DIC".*

    **Author's response:** Thank you for pointing out this typographical error.

    **Changes to the manuscript:** We have corrected these in Table 1 to "N.d.".

**R2.C2: "***the note below Table 1 for "Sp. Cond" does not match the values in the table.***"**

    **Author's response:** We are a little confused by this comment. If this refers to "Sp. Cond." in the table appearing as "Sp. Cond" in the legend, we will add a period to the latter in the revision.

    **Changes to the manuscript:** On Table 1 captions, "Sp. Cond" has been modified to "Sp. Cond."

**R2.C3: "***Table 2 contains two variations of "Not applicable", mentioned as "N.a" or "N.a."***".**

    **Author's response:** Thank you for pointing out this discrepancy.

**Changes to the manuscript:** We have corrected "N.a." in Table 2.

**R2.C4: "***The figures appear blurry, and the color of the graph's axes could be made darker to enhance clarity for readers.***"**

    **Author's response:** We agree with this suggestion and increased resolution for the figure images.

    **Changes to the manuscript:** Following the publisher guidelines, each figure for the final manuscript will be submitted as an individual vector image in .pdf, which will ensure they will be displayed at maximum possible resolution.

**R2.C5: "***In line 351-352, please provide accurate data or a figure to support the statement regarding "ATP-based biomass concentrations in samples from River Rise at low flow being significantly higher than values at high flow and approximately 3-fold lower than those observed for River Sink at low flow."***".**

    **Author's response:** In the initial submission, we neglected to point out that this statement is supported by the data in Fig. 3 c.

**Changes to the manuscript:** Fig. 3 c is now reference in L364 to provide support for this statement.

**R2.C6:** *Line 357-359 refers to "Fig. 2c," but there is no such figure mentioned in the article.*

    **Author's response:** Thank you for pointing out this oversight.

    **Changes to the manuscript:** Reference to Fig. 2 c in L371 has been removed.

**R2.C7:** "*In line 390-392 and line 407-408, the authors state that the incorporation rate of 3H-thymidine was much higher from Madison Blue Spring. Therefore, it may be more appropriate to use the 3H-thymidine incorporation rate when evaluating doubling times at Madison Blue Spring.*".

**Author's response:** More rapid generation times were inferred from the $^3$H-thymidine at all sites, but for Madison Blue Spring, they were much more rapid (7 to 30 h versus 232 to 1028 h for the $^3$H-leucine incorporation data; Barry Sosa, PhD thesis, University of Florida, 2023). Additionally, we observed very low ratios of leucine to thymidine incorporation at Madison Blue Spring, which were ~10-fold lower than those observed at the other sites (Table 2). This coupled with low BGE values implied uncoupling between catabolism and anabolism and/or that the community was investing most of its energy flow in maintenance metabolism rather than growth (see discussion in lines 575-578). For these reasons and those described in lines 403-405, we believe the $^3$H-leucine data are the most appropriate for estimating doubling times and bacterial production to make comparisons among sites. Please note that $^3$H-thymidine incorporation rates are provided in the supplementary materials (Figure S8), which allow anyone seeking to make direct comparisons to calculate bacterial production and doubling rates from these data.

**Changes to the manuscript:** None.

**R2.C8:** "*In line 460, please attempt to explain why ATP contents are high at Devil's Eye Spring.*"

**Author's response:** The very high per cell ATP contents inferred in groundwater from Devil's Eye Spring are surprising and remain unexplained. The simplest possible explanations are that 1) the groundwater community at Devil's Eye has higher amounts of biomass/ATP, 2) is producing ATP relatively more rapidly than the communities at other sites, or 3) the excesses are due to low anabolic rates that consume ATP slowly and allow produced ATP to accumulate in the cells. Given that the first two options are not supported by our data (see lines 462-464), the latter explanation (#3) is the most likely.

**Changes to the manuscript:** In L486-487, we added text that explores the possibilities described above and states this working hypothesis.